

# Late Quaternary glacial maxima in Southern Patagonia: insights from the Lago Argentino glacier lobe

Matias Romero[1,2,3], Shanti B. Penprase[4,5], Maximillian S. Van Wyk de Vries[4,5,6,7,8,9,10], Andrew D. Wickert[4,5,11], Andrew G. Jones[3], Shaun A. Marcott[3], Jorge A. Strelin[2,12], Mateo A. Martini[1,2], Tammy M. Rittenour[13], Guido Brignone[1], Mark D. Shapley[14], Emi Ito[4,14], Kelly R. MacGregor[15], and Marc W. Caffee[16,17]

[1]Facultad de Ciencias Exactas, Físicas y Naturales (FCEFyN), Universidad Nacional de Córdoba, Av. Haya de la Torre, Córdoba, X5000HUA, Argentina
[2]Centro de Investigaciones en Ciencias de la Tierra (CICTERRA), Consejo Nacional de Investigaciones Científicas y Tecnológicas (CONICET), Córdoba, X5000IND, Argentina
[3]Department of Geoscience, University of Wisconsin-Madison, Madison, Wisconsin, USA
[4]Department of Earth & Environmental Sciences, University of Minnesota, Minneapolis, MN 55455, USA
[5]Saint Anthony Falls Laboratory, University of Minnesota, Minneapolis, MN 55455, USA
[6]School of Environmental Sciences, University of Liverpool, Liverpool, L3 5DA, UK
[7]School of Geography and the Environment, University of Oxford, Oxford OX1 3QY, UK
[8]School of Geography, University of Nottingham, Nottingham, NG7 2RD, UK
[9]Department of Geography, University of Cambridge, Cambridge CB2 3EL, UK
[10]Department of Earth Sciences, University of Cambridge, Cambridge CB3 0EZ, UK
[11]Sektion 4.6: Geomorphologie, Deutsches GeoForschungsZentrum (GFZ), Potsdam, 14473, Germany
[12]Instituto Antártico Argentino, Argentina
[13]Department of Geosciences, Utah State University, Logan, UT 84322
[14]Continental Scientific Drilling Facility, Department of Earth and Environmental Sciences, University of Minnesota, Minneapolis, MN, 55455, USA
[15]Department of Geology, Macalester College, Saint Paul, Minnesota 55105, USA
[16]Department of Physics and Astronomy, Purdue University, West Lafayette, IN 47907, USA
[17]Department of Earth, Atmospheric, and Planetary Science, Purdue University, West Lafayette, IN 47907, USA

**Correspondence:** Matias Romero (mromero6@wisc.edu)

**Abstract.** Determining the timing and extent of Quaternary glaciations around the globe is critical to understanding the drivers behind climate change and glacier fluctuations. Despite synchronous ice-volume and extent change across hemispheres, evidence from the southern mid-latitudes indicates that local glacial maxima occurred earlier in the glacial cycle, preceding the global Last Glacial Maximum (LGM), implying that feedbacks in the climate system or ice dynamics played a role beyond

5    the underlying orbital parameters. To shed light on these processes, we investigated the glacial landforms shaped by the Lago Argentino glacier (50° S), an outlet lobe of the former Patagonian Ice Sheet in southern Argentina, during the last two glacial cycles. We mapped geomorphological features on the landscape and dated moraine boulders and outwash sediments using [10]Be cosmogenic nuclides and feldspar infrared stimulated luminescence (IRSL) to constrain the chronology of glacial advance and retreat. We, therefore, provide the first published age constraints on the middle-to-late Pleistocene glaciations at

10   Lago Argentino, and report that this outlet lobe expanded during Marine Isotope Stage 6, at 153.0±14.7 ka, and during Marine Isotope Stage 3, culminating at 44.5±8.0 ka and at 36.6±1.0 ka. Our results indicate that the most recent was its most





extensive advance during the last glacial period, and hypothesize that this was a result of longer and colder winters, as well as increased precipitation delivered by a northward migration of the Southern Westerly Winds belt, highlighting the role of local and regional climate feedbacks in driving ice mass changes in the southern mid-latitudes.

## 1 Introduction

Unraveling the timing of continental ice sheet growth and decay is crucial to determine glacier response to climate forcings and feedbacks within the Earth's system. However, our understanding of the climatic drivers behind global glacier change is limited by the availability of highly resolved past glacial records. For instance, Northern Hemisphere insolation intensity has been proposed to pace the climate of the Southern Hemisphere during the last million years, implying synchronous ice sheet growth and decay across hemispheres every ~100 ka, coincident with the eccentricity cycles of the Earth's orbit (Abe-Ouchi et al., 2013; Hays et al., 1976; Imbrie et al., 1993). Nonetheless, evidence indicates that glaciers in mid-to-high latitudes in the Southern Hemisphere expanded prior to the global Last Glacial Maximum (LGM, 26.5–19 ka; Clark et al., 2009), suggesting that other mechanisms, apart from the underlying orbital parameters, could have played a role in inducing pre-LGM glacier growth, such as local ice dynamics, ocean-atmosphere interactions, and latitudinal shift of the Southern Westerly Winds (SWW; Fig. 1A; Darvill et al., 2015, 2016; Denton et al., 2021; Doughty et al., 2015; García et al., 2018; Hall et al., 2020; Mendelová et al., 2020; Shulmeister et al., 2019).

Since glacial chronologies are necessary to determine the timing and occurrence of glacial expansions and recessions, dating glacial deposits provides a first order constraint on past glacier fluctuations. In South America, evidence indicates that the Patagonian Ice Sheet (PIS) expanded during the last glacial cycle (115–11.7 ka, Hughes et al., 2013), reaching its maximum extent between 35–28 ka in northern sites and around 47 ka in southern sites (Davies et al., 2020). The PIS formed a continuous ice sheet along the spine of the Andean Cordillera from 38°S to 55°S (Fig. 1B), with a sea level equivalent of approximately 1.5 m (Davies et al., 2020; Hulton et al., 2002)

Lago Argentino (Fig. 1B) is located on the eastern flank of the Southern Patagonian Icefield and constitutes the largest ice-contact lake in the world, with multiple lake-terminating glaciers calving into it (Van Wyk de Vries et al., 2022). Lago Argentino drains into the Río Santa Cruz basin and, ultimately, the Atlantic Ocean. While those moraines on the western flank of the Andes are poorly preserved given higher weathering rates due to high rainfall, the strong rain shadow effect imposed by the Andean Range (Garreaud et al., 2009) preserves landforms on its arid eastern foreland, making moraines located in the Argentine steppe more suitable for geochronological dating. Despite well-dated glacial records for the Lago Argentino and Rio Santa Cruz basin during the Plio-Pleistocene (Clague et al., 2020; Mercer, 1976; Strelin and Malagnino, 1996; Strelin et al., 1999) as well as during the late Glacial and Holocene (Kaplan et al., 2011; Strelin et al., 2011, 2014), the middle-to-late Pleistocene glacial history of the Lago Argentino glacier lobe, an outlet lobe of the former PIS, remain largely unstudied.

To fill this data gap on the age and extent of the Lago Argentino glacier lobe in the remainder of the Pleistocene and to improve the understanding of glacial cycles in the Southern Hemisphere during this period, we produce a new highly-resolved geomorphological map of the upper basin of the Río Santa Cruz, date moraine-crest boulders with cosmogenic nuclide surface



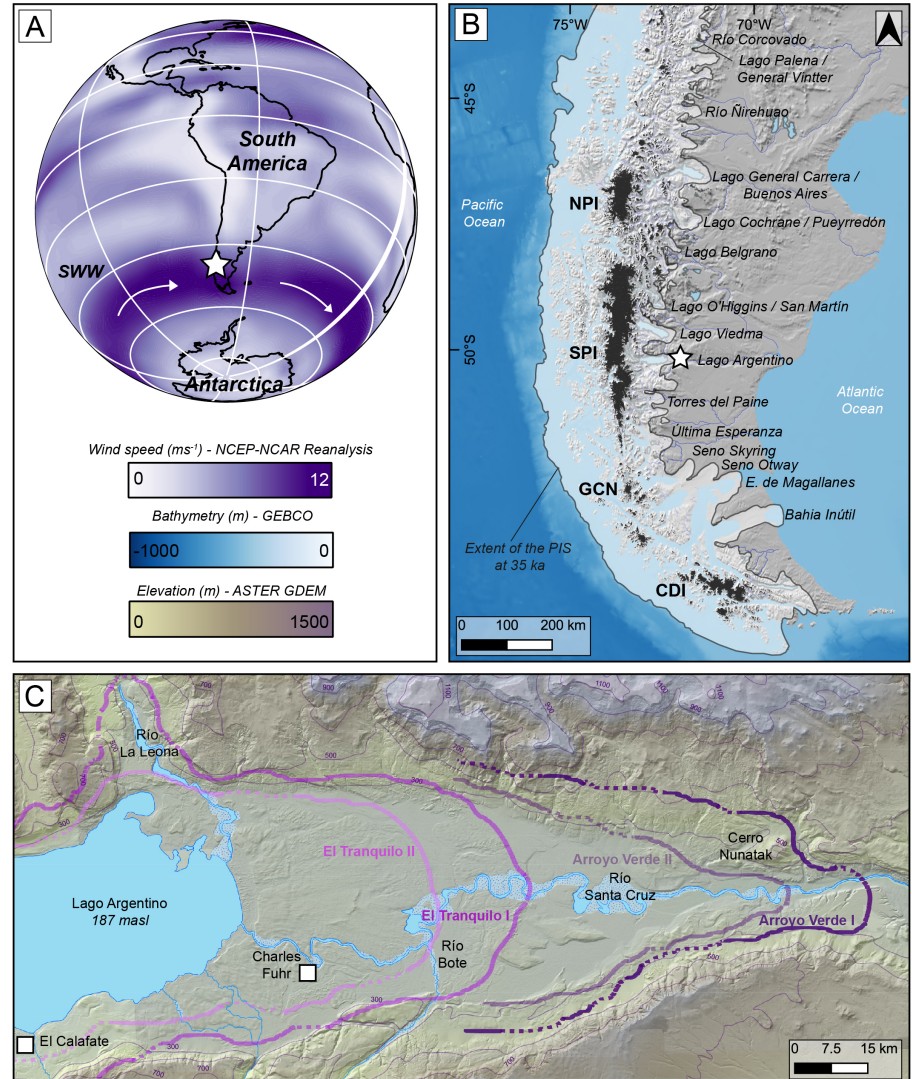

**Figure 1. (A)** Location of study site in a hemispheric context along with Southern Westerly Winds (SWW) speed data at 850 mb after Kalnay et al. (1996), **(B)** Ice extent of the former Patagonian Ice Sheet at 35 ka (Davies et al., 2020) along with major outlet lobes, topographic hillshade, and bathymetry data. The white star denotes the location of the study site (Lago Argentino glacier lobe), **(C)** Digital Elevation Model (DEM) of the upper basin of Río Santa Cruz located eastward of Lago Argentino. Shades of purple indicate the defined (solid lines) and inferred (dashed lines) of the middle-to-late Pleistocene glaciation limits (Strelin and Malagnino, 1996, 2009).

exposure dating, and date proglacial outwash using feldspar Infrared Stimulated Luminescence (IRSL). We report the extent of the Lago Argentino glacier lobe during the last two glacial cycles during Marine Isotope Stages (MIS) 6 and 3, and show that the maximum ice extent during the last glaciation occurred during MIS 3, preceding the Northern Hemisphere glacial maximum





that occurred during MIS 2. Therefore, this dataset highlights the PIS sensitivity to local and regional climate feedbacks and provides an improved understanding of the landscape evolution in the Lago Argentino basin during the Pleistocene.

## 2 Background

### 2.1 Physical setting

The Río Santa Cruz valley (∼50° S) runs ∼250 km from the easternmost end of Lago Argentino to the Atlantic Ocean. Late Miocene to early Pliocene basaltic plateaus bound it to the north and south (Ramos and Kay, 1992). These plateaus are underlain by early to middle Mesozoic marine and fluvial sedimentary successions (Casadio et al., 2000; Goyanes and Massabie, 2015). The upper sector of this valley consists of a broad plain (Fig. 1C) that narrows ∼65 km east of the lake's margin, where glacial landforms are present (Strelin and Malagnino, 1996).

The drainage basin surface area is about $30x10^3$ km$^2$, where river length is ∼385 km with a maximum modern water discharge of ∼1200 m$^3$/s in March, while minimum discharge values are measured in September (∼300 m$^3$/s, Pasquini et al., 2021). Discharge has increased in the Lago Argentino basin throughout the 21$^{st}$ century due to rapid and accelerating ice loss on the eastern flank of the icefields (Van Wyk De Vries et al., 2023). The Southern Westerly Winds (SWW) deliver the bulk of the precipitation from the west in a predominant SW-NE direction (Fig. 1A), with greater summer intensity (Garreaud et al., 2009). In a similar seasonal fashion, ice melt rates and glacier velocity reach their maximum during the summer (Minowa et al., 2017, 2021; Mouginot and Rignot, 2015). Precipitation is abundant on the west of the Andean range and reaches annual values of between 5000-10000 mm, but the semiarid terrain around Lago Argentino receives less than 200 mm per year (Garreaud et al., 2013; Lenaerts et al., 2014).

### 2.2 Glacial history and previous studies

Several authors have contributed to the understanding of the local landscape evolution, even though no chronological evidence is available for the middle-to-late Pleistocene glacial cycles. During his journey to South America, Darwin (1842) discussed the possible glaciomarine origin of erratic boulders in the Río Santa Cruz basin. The first geomorphological assessment was carried out by Caldenius (1932), who identified four distinct moraine belts, and according to their preservation state, assigned a relative ages based on the Fennoscandian glaciations (De Geer, 1927). Caldenius (1932) named these moraine systems (from eastern outer to western inner) Initioglacial, Daniglacial, Gotiglacial, Finiglacial, with the latter corresponding to the Puerto Banderas moraines. These moraines are known to date from the Antarctic Cold Reversal, deposited about 10 km from the modern ice from at ∼13,000 cal yrs before present (Strelin et al., 2011).

Based on this framework and new observations, Feruglio (1944) later grouped the Daniglacial and Gotiglacial limits from the upper Río Santa Cruz Valley and assigned them to the Last Glaciation. The observations made by Feruglio (1944) were later revisited by several authors, who ultimately assigned Caldenius' Finiglacial moraines to the Last Glaciation, and the outer moraines systems to the early and mid Pleistocene glaciations (Mercer, 1976; Strelin and Malagnino, 1996; Strelin et al.,





1999; Strelin and Malagnino, 2009; Wenzens, 1999, 2005; Rabassa and Clapperton, 1990). Mercer (1976) provided one of the

first geochronological constraints by dating lava flows overlying glacial deposits, and determined the easternmost glaciations occurred during the Pliocene and early Pleistocene at 3.5–1.5 Ma. These results were supported by additional dating carried out in the vicinity of Lago Viedma and Lago Argentino (Clague et al., 2020).

Strelin and Malagnino (1996, 2009) mapped and described glacial deposits and provided the first geomorphological map with its associated stratigraphy. They proposed that five glaciations occurred in the upper Río Santa Cruz basin and named them

(from eastern outer to western inner) Estancia La Fructuosa, Chuñi Aike, Cerro Fortaleza, Arroyo Verde, and El Tranquilo. For this work, we only focus on the last two glaciations identified by Strelin and Malagnino (1996): the Arroyo Verde and El Tranquilo moraines (Fig. 1C). These authors interpreted that the Lago Argentino glacier lobe deposited the Arroyo Verde I and II moraines at one of the narrowest points of the valley, and then El Tranquilo I and II moraines, located closer to the lake's margin (Fig. 1C). Based on a morphostratigraphic approach, they suggested that the Arroyo Verde I and II moraines were

deposited during the Penultimate Glaciation, Marine Isotope Stage 6 (MIS 6, 191–130 ka; Lisiecki and Raymo, 2005), and that a large proglacial lake developed after the glacier receded. Later, they suggested that El Tranquilo I and II moraines could have been deposited during two different stadials throughout the last glacial cycle, but without the geochronology necessary to confirm this.

## 3    Methods

### 3.1    Geomorphological mapping

We built on previous mapping efforts and geomorphological studies carried out in the vicinity of the upper basin of Río Santa Cruz to create a new, detailed geomorphological map of the Arroyo Verde and El Tranquilo glaciations over an areal extent of ~2000 km$^2$ (Strelin and Malagnino, 1996; Strelin et al., 1999; Strelin and Malagnino, 2009). We identified landforms remotely using a 5 m resolution aerial-photogrammetry-derived digital elevation model (DEM) provided by the Instituto Ge-

ografico Nacional of Argentina (IGN) and an orthomosaic of the area (pixel = 40 cm). We complemented these using Google Earth satellite imagery and ASTER GDEM. We conducted field validation during 2019 and 2020 to complement the remote sensing and verify initial geomorphological interpretations (Chandler et al., 2018). We assessed glacial landsystems by performing sedimentary and stratigraphic loggings around the major landforms, and then manually digitized the geomorphological features using ArcMap 10.5 and QGIS, where we created both polygons and polylines to delineate the landforms. We followed

previously published criteria (Cooper et al., 2021; Leger et al., 2020; Mendelová et al., 2020; Peltier et al., 2023; Soteres et al., 2020, 2022), and we classified these features based on their primary depositional environment as ice marginal (e.g., moraines, hummocks), subglacial (e.g., glacial lineations, drumlins), glaciofluvial (e.g., outwash plains), glaciolacustrine, and other non-glacial features (such as modern hydrography). Lastly, we used built-in geospatial tools to derive the count, length, and orientation of the mapped topographic features.



## 3.2 Geochronological dating

### 3.2.1 $^{10}$Be cosmogenic nuclide surface exposure dating

We determine the age of moraine abandonment (therefore, culmination of ice advance) by using $^{10}$Be cosmogenic nuclide surface exposure dating on moraine boulders. We targeted boulders from the Arroyo Verde and El Tranquilo moraines as defined by Strelin and Malagnino (1996). We sampled eighteen quartz-bearing boulders of 0.5–3 m in height, deposited across three different moraine complexes (i.e., Arroyo Verde II, El Tranquilo I, El Tranquilo II). We targeted subrounded boulders lacking erosional features (e.g., pitting) to avoid removal of nuclides due to post-depositional processes, and sampled those exposed on moraine crests. We collected approximately 1.5–2 kg from flat surfaces using a hammer and a chisel. We report the location (latitude, longitude, elevation) of the sampled boulders using a handheld GPS with a vertical uncertainty of < 10 m. We account for topographic shielding using a GIS-based toolbox (Li, 2013, 2018) on the DEMs employed for mapping (see section 3.1). We provide field and analytical information in Table 1.

We processed the samples for $^{10}$Be extraction at the University of Wisconsin-Madison following standard laboratory procedures (Ceperley et al., 2020; Jones et al., 2023), using a $^9$Be carrier solution prepared from raw beryl (OSU White, $^9$Be concentration = 251.6±0.9 ppm; Marcott, 2011). $^{10}$Be/$^9$Be ratios were measured at the Purdue University Rare Isotope Measurement Laboratory (PRIME Lab) and normalized to standard 07KNSTD3110, which has an assumed $^{10}$Be/$^9$Be ratio of $2.85 \times 10^{12}$ (Nishiizumi et al., 2007). Lastly, we background-corrected the $^{10}$Be concentrations with batch-specific blank values (Table 1, S2).

We calculated exposure ages using Version 3 of the CRONUS-Earth online calculator (last accessed October 20th, 2023 Balco et al., 2008), employing a local production rate developed for the late Glacial chronology at Lago Argentino (3.71± 0.11 atoms/g/yr; Kaplan et al., 2011). We did not apply snow correction to our ages given the low precipitation levels in the area today, and we considered a rock density of 2.65 g/cm$^3$. As part of a sensitivity test, we calculated the ages with different erosion rates for all landforms, ranging from 0.2–1.4 mm/ka according to Douglass et al. (2006) and Kaplan et al. (2005), respectively. Since the outcomes of using different erosion rates (Table S1, Fig. S6) do not change the main results of this work and given that age differences overlap within analytical uncertainties, we assumed the erosion rate to be zero for all the samples for our reported ages and interpretations. Here, we report the calculated ages using the time dependent scaling (Lm: Lal, 1991; Stone, 2000), the non-time-dependent scaling (St: Stone, 2000; Lal, 1991), and the LSDn scaling scheme developed by Lifton et al. (2014). Even though we account for different scaling schemes listed in Table 2, the choice of the scaling scheme (LSDn) does not impact our interpretations. Lastly, we employ the iceTEA toolbox to plot age distribution and to identify outliers (Jones et al., 2019).

### 3.2.2 Feldspar infrared stimulated luminescence (IRSL) dating

We collected four proglacial sediment samples and two loess samples to date the timing of outwash and aeolian deposition using infrared stimulated luminescence (IRSL, Table 3). Luminescence approaches, including optically stimulated luminescence (OSL) and IRSL are well-suited to date the timing of deposition and subsequent burial of sediment grains. While other studies





**Figure 2. (A)** Quaternary geomorphological map of the main glacial landforms in the upper Río Santa Cruz basin (Fig. 1C): ice marginal (moraines, hummocks), subglacial (drumlins, lineations), glaciolacustrine, glaciofluvial (outwash plains), as well as other features. Panels **(B–F)** are close views of the mapped landforms along with geochronological results denoting the age in thousands of years (kilo annum, ka) with $1\sigma$ analytical uncertainty of the boulder sampled (yellow circles). Luminescence samples (orange circles) are associated with their corresponding profile on Fig. 5.





in Patagonia have been able to use quartz grains and OSL dating in glaciogenic sediments (Smedley et al., 2016), due to poor quartz quality and paucity in the local lithology of Río Santa Cruz, we used feldspar grains and IRSL to date these sediments.

We carried out sample collection according to established field procedures published by Nelson et al. (2015) using opaque aluminum tubes, opaque plastic caps, and a rubber mallet. We processed the samples at Utah State University Luminescence Laboratory following standard procedures involving sieving to 150-250 $\mu$m, HCl and $H_2O_2$ treatments to remove carbonates and organics, heavy mineral separation at 2.58 g/cm$^3$ with no HF pre-treatment, to isolate the potassium-rich feldspar component. The IRSL signal was measured on 1-2 mm aliquots ($\sim$20-50 grains) at 50°C according to the single-aliquot regenerative-dose

(SAR) procedures for potassium feldspar dating (Wallinga et al., 2000). Measurements were performed on Risø TL/OSL Model DA-20 readers with infrared light-emitting diodes (LEDs) (870$\pm$40 nm, $\sim$ 120 mW/cm$^2$) and signals were detected through a blue filter pack of 2-mm and 4-mm thick filters (BG-39 and Corning 7-39, respectively). IRSL ages were corrected for fading (loss of signal with time, Huntley and Lamothe, 2001; Auclair et al., 2003) and reported using the central and minimum age models (CAM and MAM, Galbraith and Roberts, 2012) based on skew in the aliquot data indicative of partial bleaching (see

figure S7 in supplemental data for radial plots). Samples for dose-rate determination were collected in sediments surrounding the sample tube and analyzed for elemental concentration using ICP-MS and ICP-AES techniques. These concentration values were converted to dose rate following the conversion factors of Guérin et al. (2011) and beta attenuation values of Brennan (2003) using the DRAC on-line dose-rate calculator (Durcan et al., 2015). The beta dose included contribution from 12.5% internal potassium and 400 ppm Rb and an a-value of 0.086. Contribution of cosmic radiation to the dose rate was calculated

using sample depth, elevation and latitude/longitude following Prescott and Hutton (1994). Total dose rates were calculated based on water content, radioisotope concentrations, and cosmic contribution Table 4.

## 4 Results

### 4.1 Geomorphological mapping

#### 4.1.1 Ice marginal features - moraine complexes

We mapped and defined moraine complexes as confined areas of positive relief with curved and continuous shapes composed by moraine ridges and hummocks (Leger et al., 2020) that are separated from each other by outwash plains (Peltier et al., 2023). Moraine ridges are linear-to-curved high-relief glacigenic landforms that exhibit a break in slope, while hummocks are subrounded to rounded mounds (Leger et al., 2020). We identified four major moraine complexes in the upper Río Santa Cruz valley (Fig. 2, S1, S2): Arroyo Verde I, Arroyo Verde II, El Tranquilo I and El Tranquilo II that indicate the former position of

the Lago Argentino glacier lobe (Strelin and Malagnino, 1996, 2009).

The Arroyo Verde I moraines are located at one of the easternmost edges of the upper basin of the Río Santa Cruz, where the valley reaches its narrowest point, following a discontinuous semicircular pattern. Numerous moraine crests reach elevations between 270 and 350 m in the southeastern margin, while they can be recognized in the northeastern margin up to 450 m at the flanks of Cerro Nunatak, and about 500 m in the northwest, close to Río La Leona basin (Fig. 1C). The moraines are less



**Table 1.** Geographical and analytical data for the [10]Be exposure age calculation of the moraine complexes deposited by the Lago Argentino glacier lobe.

| Sample | Latitude (DD) | Longitude (DD) | Elevation (masl) | Thickness (cm) | Shielding | Boulder height (cm) | Quartz weight (g) | [9]Be carrier (g) | [10]Be/[9]Be (10⁻¹⁵) ± uncertainty | [10]Be concentration[1] (10⁵ atoms g⁻¹) ± uncertainty |
|---|---|---|---|---|---|---|---|---|---|---|
| **Arroyo Verde II** | | | | | | | | | | |
| AV-01 | -50.2972 | -71.5469 | 288 | 1.9 | 0.9980 | 3.5 | 14.3784 | 0.7697 | 1481.24±18.54 | 13.32±5.93[a] |
| AV-02 | -50.2437 | -71.3764 | 264 | 3.3 | 0.9988 | 0.5 | 24.1632 | 0.7710 | 1337.44±21.58 | 7.17±3.27[a] |
| AV-03 | -50.1995 | -71.4327 | 244 | 1.1 | 0.9979 | 0.5 | 17.5606 | 0.7673 | 1196.55±20.62 | 8.78±4.05[b] |
| AV-04 | -50.1888 | -71.4100 | 293 | 1.6 | 0.9993 | 1.5 | 13.5945 | 0.7660 | 965.76±18.14 | 9.14±4.27[b] |
| **El Tranquilo I** | | | | | | | | | | |
| ET-02 | -50.2906 | -71.7018 | 222 | 1.6 | 0.9994 | 0.4 | 12.0819 | 0.7707 | 293.94±7.27 | 3.14±1.55[a] |
| ET-04 | -50.2822 | -71.6391 | 223 | 3.2 | 0.9993 | 0.4 | 5.0401 | 0.7683 | 71.53±3.67 | 1.81±1.23[b] |
| ET-06 | -50.2638 | -71.6294 | 269 | 1.5 | 0.9979 | 0.6 | 4.2382 | 0.7707 | 57.48±2.93 | 1.73±1.17[a] |
| ET-12 | -50.1415 | -71.6923 | 226 | 3.6 | 0.9998 | 0.5 | 20.4503 | 0.7689 | 358.75±7.66 | 2.26±1.08[a] |
| ET-13 | -50.1363 | -71.6979 | 228 | 0.9 | 0.9975 | 1.6 | 3.6626 | 0.7645 | 60.18±8.70 | 2.06±3.20[c] |
| ET-14 | -50.1299 | -71.6762 | 279 | 1.5 | 0.9970 | 3.5 | 11.5055 | 0.7691 | 230.18±9.51 | 2.58±1.54[a] |
| ET-17 | -50.1673 | -71.6382 | 208 | 2.2 | 0.9998 | 0.6 | 5.998 | 0.7680 | 92.20±4.12 | 1.97±1.23[b] |
| ET-18 | -50.1702 | -71.6491 | 217 | 2.2 | 0.9986 | 0.6 | 21.4128 | 0.7681 | 243.15±8.43 | 1.46±0.81[b] |
| **El Tranquilo II** | | | | | | | | | | |
| ET-07 | -50.1090 | -72.0998 | 262 | 8.4 | 0.9916 | 1.5 | 21.2531 | 0.7686 | 322.94±8.07 | 1.96±0.97[b] |
| ET-08 | -50.1111 | -72.1079 | 262 | 3.7 | 0.9985 | 1.5 | 18.8957 | 0.7724 | 294.77±7.69 | 2.02±1.01[a] |
| ET-09 | -50.0996 | -72.1191 | 285 | 2.1 | 0.9996 | 1.5 | 20.7598 | 0.7687 | 322.98±8.05 | 2.00±0.99[b] |
| ET-10 | -50.1090 | -72.0817 | 275 | 4.0 | 0.9995 | 2.5 | 19.8971 | 0.7702 | 319.28±6.92 | 2.07±0.99[a] |
| ET-11 | -50.1038 | -72.0813 | 287 | 8.1 | 0.9987 | 1.5 | 8.2608 | 0.7671 | 121.54±5.27 | 1.88±1.15[b] |
| LA-01 | -50.3138 | -72.1261 | 255 | 2.3 | 0.9988 | 5.0 | 22.1544 | 0.7656 | 347.45±19.59 | 2.01±1.43[c] |

[1] [10]Be concentrations have been background-corrected from the mean of their process-blank [10]Be/[9]Be ratios:

a) 1.06, 0.62, 1.24 x 10⁻¹⁵, b) 0.77, 0.69, 0.95 x 10⁻¹⁵, and c) 2.33, 0.68 x 10⁻¹⁵.

well-preserved on the valley slopes, where mass wasting deposits are widespread. The Arroyo Verde II moraines are located west of the Arroyo Verde I moraines and comprise multiple ridges and low-relief hummocks (∼10 m).

El Tranquilo I moraines are located about 20 km to the west of the Arroyo Verde II moraines, and form a quasi-continuous and semicircular pattern comprised of several moraine ridges and abundant low-relief hummocks (5–10 m) in the frontal portion. These deposits are well-developed and reach elevations up to 400 m in the northern margin, compared to less-developed moraines that reach elevations below 300 m in the southern margin. Lastly, El Tranquilo II moraines are located ∼10 km inboard of the El Tranquilo I moraines, separated by an extensive outwash plain in the central portion of the valley. Here, these



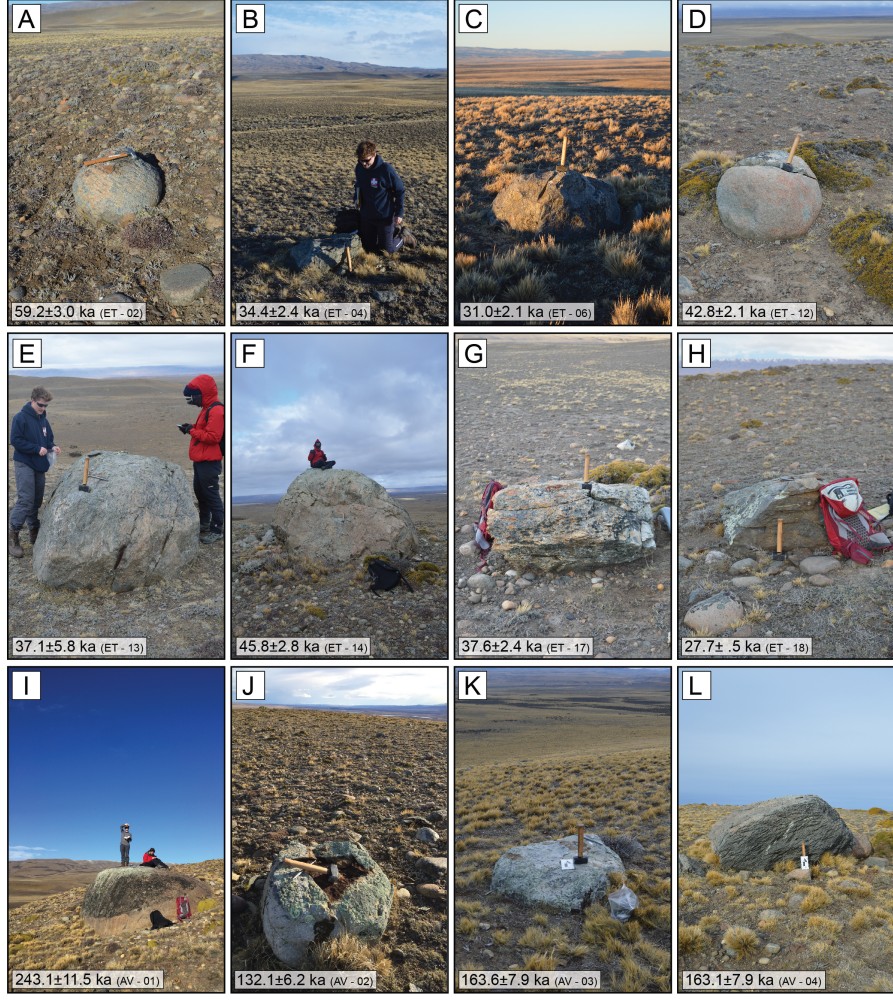

**Figure 3.** Field photos of the sampled boulders for **(A–H)** El Tranquilo I and **(I–L)** Arroyo Verde II moraines and their corresponding $^{10}$Be exposure ages in thousands of years (ka) with $1\sigma$ analytical uncertainty.

moraines exhibit low-gradient and low-relief surfaces and are intersected by alluvial fans and paleochannels. They are found partially covered by an extensive aeolian sediments field. This moraine complex is better developed at the northern margin of the Lago Argentino, where several lateral ridges are evident, but it is less-preserved in the southern margin.

### 4.1.2 Subglacial features

We defined drumlinized terrains as diffused areas that contain linear features indicative of ice flow direction (Soteres et al., 2020). We identified and mapped over 600 subglacial features, including drumlins (linear depositional landforms with oval or ellipsoidal shapes) and glacial lineations (linear and narrow depositional landforms) inboard the main moraine complexes





**Table 2.** Cosmogenic nuclide surface exposure ages of moraine boulders according to the different scaling schemes; the LSDn scaling of Lifton et al. (2014) in bold, the non-time-dependent scaling (St) from Stone (2000)/Lal (1991), and the time dependent scaling (Lm) of Stone (2000)/Lal (1991) along with $1\sigma$ internal and external uncertainties. Mean ages in bold report the mean age of the moraine complex after removing outliers (indicated with *)

| Sample | LSDn | | | St | | | Lm | | |
|---|---|---|---|---|---|---|---|---|---|
| | Age (ka) | Int (ka) | Ext (ka) | Age (ka) | Int (ka) | Ext (ka) | Age (ka) | Int (ka) | Ext (ka) |
| **Arroyo Verde II** | | | | | | | | | |
| Mean age: 175.5±47.4 ka (n=4) | | | | | | | | | |
| **Mean age: 153.0±14.7 ka (n=3)** | | | | | | | | | |
| | | | | | | | | | |
| AV-01* | **243.1** | 11.5 | 14.2 | 258.1 | 12.3 | 15.4 | 250.1 | 11.9 | 14.8 |
| AV-02 | **132.1** | 6.2 | 7.7 | 139.4 | 6.6 | 8.2 | 135.6 | 6.4 | 7.9 |
| AV-03 | **163.6** | 7.9 | 9.6 | 172.6 | 8.3 | 10.3 | 167.8 | 8.1 | 10.0 |
| AV-04 | **163.1** | 7.9 | 9.7 | 172.1 | 8.4 | 10.4 | 167.2 | 8.1 | 10.0 |
| **El Tranquilo I** | | | | | | | | | |
| Mean age: 39.4±9.9 ka (n=8) | | | | | | | | | |
| **Mean age: 44.5±8.0 ka (n=5)** | | | | | | | | | |
| | | | | | | | | | |
| ET-02 | **59.2** | 3.0 | 3.6 | 61.5 | 3.1 | 3.7 | 60.2 | 3.0 | 3.7 |
| ET-04* | **34.4** | 2.4 | 2.6 | 35.7 | 2.4 | 2.7 | 35.0 | 2.5 | 2.7 |
| ET-06* | **31.0** | 2.1 | 2.4 | 32.1 | 2.2 | 2.5 | 31.5 | 2.2 | 2.4 |
| ET-12 | **42.8** | 2.1 | 2.5 | 44.7 | 2.2 | 2.7 | 43.7 | 2.1 | 2.6 |
| ET-13 | **37.1** | 5.8 | 6.0 | 38.5 | 6.0 | 6.2 | 37.7 | 5.9 | 6.1 |
| ET-14 | **45.8** | 2.8 | 3.2 | 47.8 | 2.9 | 3.3 | 46.8 | 2.8 | 3.2 |
| ET-17 | **37.6** | 2.4 | 2.7 | 39.0 | 2.5 | 2.8 | 38.2 | 2.4 | 2.7 |
| ET-18* | **27.7** | 1.5 | 1.8 | 28.7 | 1.6 | 1.9 | 28.2 | 1.6 | 1.8 |
| **El Tranquilo II** | | | | | | | | | |
| **Mean age: 36.6±1.0 ka (n=6)** | | | | | | | | | |
| | | | | | | | | | |
| ET-07 | **37.8** | 1.9 | 2.3 | 39.2 | 2.0 | 2.4 | 38.4 | 1.9 | 2.3 |
| ET-08 | **37.2** | 1.9 | 2.3 | 38.6 | 2.0 | 2.4 | 37.9 | 1.9 | 2.3 |
| ET-09 | **35.7** | 1.8 | 2.1 | 37.0 | 1.8 | 2.2 | 36.3 | 1.8 | 2.1 |
| ET-10 | **37.8** | 1.8 | 2.2 | 39.2 | 1.9 | 2.3 | 38.4 | 1.9 | 2.3 |
| ET-11 | **35.2** | 2.2 | 2.5 | 36.5 | 2.3 | 2.6 | 35.8 | 2.3 | 2.5 |
| LA-01 | **35.8** | 2.6 | 2.8 | 37.1 | 2.7 | 3.0 | 36.4 | 2.6 | 2.9 |





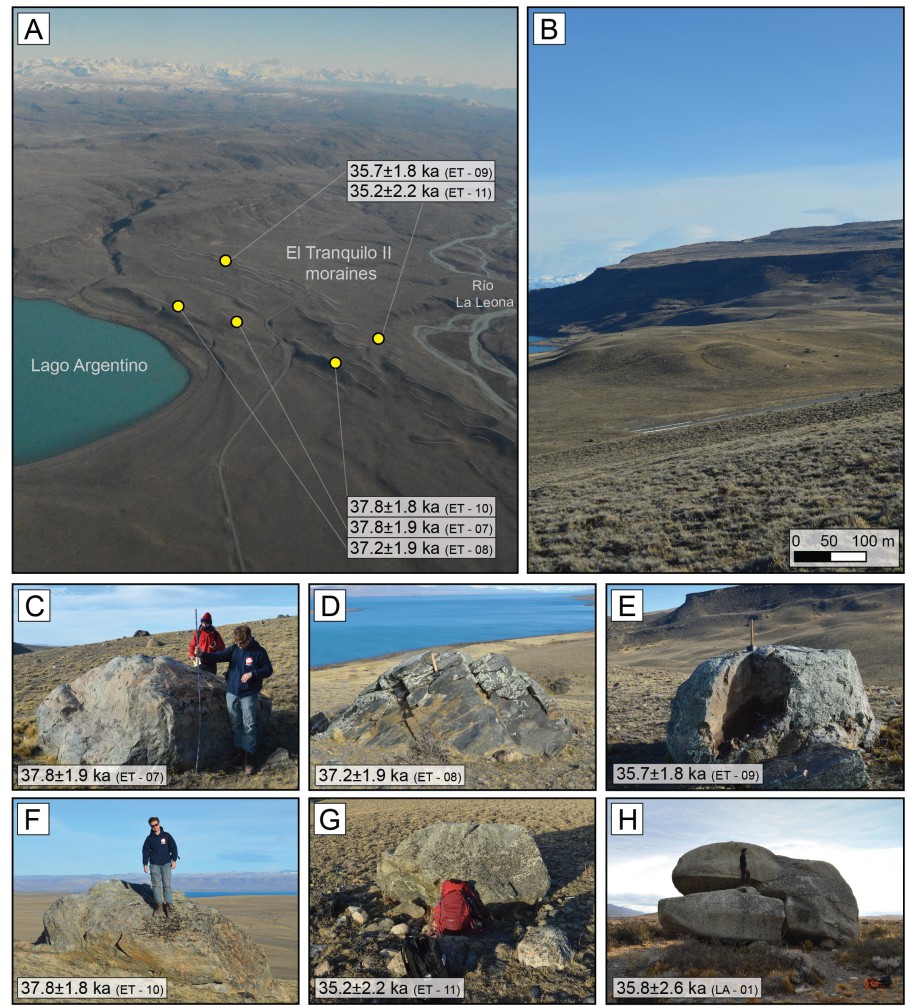

**Figure 4. (A)** Aerial view of the El Tranquilo II moraines located at the northeastern margin of Lago Argentino along with the $^{10}$Be exposure ages obtained for the sampled boulders in thousands of years (ka) with $1\sigma$ analytical uncertainty. **(B)** On-land view of the El Tranquilo II moraines shown in panel A. **(C–H)** boulders sampled from this moraine system. Note that sample LA-01 **(H)** is not pictured in the aerial photo, but is part of the same moraine system.

(Leger et al., 2020), with a mean length of 372 m (Fig. 2, S2, S3). The drumlin field is extensive in the center of the valley,
particularly inboard of the El Tranquilo II moraines, where it exhibits a fan-like distribution parallel to the direction of ice flow, with a predominantly W–E orientation for the longest axis and an additional NW–SE component.



### 4.1.3 Glaciofluvial features

We identified outwash plains as extensive low-gradient proglacial surfaces composed of sands and rounded gravels (Soteres et al., 2020), and associated them with their corresponding moraine complexes, following the framework established by in Strelin and Malagnino (1996). While the Arroyo Verde glaciofluvials are located at eastern sites and at higher elevations, the El Tranquilo outwash plains are widespread in the center of the valley at elevations between 180 – 200 m with low-gradient slopes (∼0.3 to 1 %). Notably, the El Tranquilo I outwash is located outboard of its corresponding moraine system, whereas the El Tranquilo II outwash is found outboard of both moraine complexes. Strelin and Malagnino (1996) indicated that El Tranquilo I outwash was incised by glacial meltwater and later re-filled by El Tranquilo II outwash. Additionally, abundant paleochannels can be traced in the southern margin of the valley, following a braided pattern, as noticeable in the satellite imagery (Fig. 2, S3).

### 4.1.4 Glaciolacustrine features

Glaciolacustrine landforms and glacial transgressive facies are mainly found in the center of the valley, outboard of El Tranquilo I moraines, which partially cover the surface of the glaciolacustrine deposits (Fig. 2). Exposed sections show deformed silt-sized sequences with thicknesses reaching up to ∼3 m or more. Lacustrine landforms are evident in the southeastern margin of the valley in the form of paleolake shorelines, paleodeltas, and spits. The northeastern margin of the valley also exhibits lacustrine landforms, which are partially covered by moraine ridges and hummocks from the Arroyo Verde II advance. However, these are found in a lesser state of preservation compared to the ones located in the center of the valley.

### 4.1.5 Other features

We identified additional features apart from modern hydrography (major rivers, rivers, lake, fluvial landscape). Evidence of former meltwater routing was mapped as paleochannels given their sinuosity, and negative relief as well as lack of modern drainage (Leger et al., 2020). Additionally, we mapped raised (paleo)lacustrine deposits and their (paleo)lake shorelines as continuous landforms with linear and parallel features circling lacustrine deposits that trace former lake level surface (Leger et al., 2020). These deposits and features are found on the eastern margin of the lake and in the central part of the valley (Strelin and Malagnino, 1996). We recognized alluvial fans based on their fan-like shape, while gravitational deposits were identified based on their bowl-like shape and break in slope (Leger et al., 2020). On the other hand, we mapped scarp surfaces according to their continuous break-in-slope and the sharp elevation change (Leger et al., 2020). Lastly, aeolian deposits were identified across the landscape during ground-validating fieldwork and mapped on the imagery based on their distinctive lighter colors as well as their elongated and parallel surface morphology.





**Table 3.** Infrared Stimulated Luminescence (IRSL) Age Information

| Sample | USU num. | Num. of aliquots[1] | Dose rate (Gy/kyr) | Fading Rate $g_{2\,days}$ (%/decade) | Age Model[2] | Equivalent Dose ± 2σ (Gy) | IRSL age[3] ± 1σ (ka) | Type |
|---|---|---|---|---|---|---|---|---|
| SCR20-OSL-01 | USU-3764 | 16 (19) | 3.0±0.1 | 5.3±0.9 | MAM | 80.9±13.3 | **50.5±6.9** | Outwash |
| SCR20-OSL-02 | USU-3765 | 13 (21) | 3.1±0.1 | 3.3±1.1 | MAM | 52.7±17.8 | **32.1±5.5** | Outwash |
| SCR20-OSL-03 | USU-3766 | 11 (15) | 3.3±0.1 | 5.2±1.0 | MAM | 52.7±15.4 | **29.2±4.8** | Outwash |
| SCR20-OSL-04 | USU-3767 | 16 (17) | 3.2±0.1 | 4.0±0.1 | CAM | 70.8±12.4 | **34.4±4.2** | Outwash |
| SCR20-OSL-05 | USU-3768 | 13 (13) | 3.0±0.1 | 4.6±0.3 | CAM | 20.9±3.6 | **10.8±1.3** | Loess |
| SCR20-OSL-06 | USU-3769 | 11 (14) | 3.2±0.1 | 4.3±1.1 | CAM | 8.1±1.1 | **3.6±0.4** | Loess |

[1] Age analysis using the single-aliquot regenerative-dose procedure of Wallinga et al. (2000) on 1-2mm small-aliquots of feldspar sand (150-250 $\mu$m) at 50°C IRSL. Number of aliquots used in age calculation and number of aliquots analyzed in parentheses.

[2] Equivalent dose (DE) and IRSL age calculated using the Central Age Model (CAM) or Minimum Age Model (MAM) of Galbraith and Roberts (2012).

[3] IRSL age on each aliquot corrected for fading following the method by Auclair et al. (2003) correction model of Huntley and Lamothe (2001).

## 4.2 Geochronological dating

### 4.2.1 Arroyo Verde moraines

We sampled four boulders for [10]Be exposure dating from the Arroyo Verde II moraine complex (Fig. 3I–L). The lateral moraines are found at both the southern and northern margin of the Río Santa Cruz Valley, and constitute the outermost moraine sampled (Fig. 2). Ages range from 243.1±11.5 ka (AV-01) to 132.1±6.24 ka (AV-02) in the southern margin, while samples collected at the northern margin resulted in 163.6±7.85 ka (AV-03) and 163.15±7.9 ka (AV-04).

### 4.2.2 El Tranquilo moraines

We report a total of fourteen [10]Be cosmogenic nuclide ages from both the outermost and innermost landforms from this moraine complex, El Tranquilo I and El Tranquilo II, respectively (Fig. 2). The eight boulders from El Tranquilo I moraines have ages ranging from 59.2 to 27.7 ka (Fig. 3A–H). Six samples from El Tranquilo II moraines yield a tightly clustered distribution of exposure ages, where the age range is 37.80–35.19 ka.

We sampled sections of the outwash plains for IRSL dating, obtaining four IRSL samples from El Tranquilo II outwash (Fig. 2, 5A–C). The western sample (Fig. 5A) was collected from a sandy layer within a 1–2 m thick unit of pebble and cobble sized gravels and provided an IRSL age of 29.2±4.8 ka (SCR20-OSL-03). Two samples were collected in the central portion of the valley, from an actively eroding river bank, where a lower unit of fine to coarse sands alternates with fine silts and provides an IRSL age of 50.5±6.9 ka (SCR20-OSL-01). The upper part of this section gives way to a ∼3.5 m thick pebble to cobble size outwash deposit that has an IRSL age of 32.1±5.5 ka (SCR20-OSL-02), which is ultimately capped by ∼0.5 m thick cobbles and silty aeolian deposits (Fig. 5B). The eastern section is a 1-m thick profile located at the southern margin of the valley (Fig. 5C). The lower part of this section is composed of coarse sands that alternate with pebble size gravels that resulted in an IRSL





**Figure 5.** Field photos of outwash sediments sampled for IRSL retrieved from the El Tranquilo II outwash plains, along with their sedimentary profiles as follows: **(A)** outwash profile where the westernmost sample was collected, **(B)** aggradational sequence at the river bank where samples SCR20-OSL-01 and 02 were collected, and **(C)** easternmost section surveyed with sampled outwash and loess deposits.





**Table 4.** Dose Rate Information

| Sample | USU num. | Depth (m) | In-situ H2O (%)[1] | Subsample fraction[2] | K (%)[3] | Rb (ppm)[3] | Th (ppm)[3] | U (ppm)[3] | Cosmic (Gy/kyr) |
|---|---|---|---|---|---|---|---|---|---|
| SCR20-OSL-01 | USU-3764 | 10.60 | 17.3 | F: 100% | 1.38±0.03 | 64.3±2.6 | 6.6±0.6 | 1.5±0.1 | 0.06±0.01 |
| SCR20-OSL-02 | USU-3765 | 2.70 | 1.0 | F: 30% | 1.47±0.04 | 65.4±2.6 | 6.1±0.6 | 1.3±0.1 | 0.14±0.01 |
|  |  |  |  | M: 55% | 1.37±0.03 | 57.4±2.3 | 5.6±0.5 | 1.2±0.1 |  |
|  |  |  |  | C: 15% | 1.95±0.05 | 91.8±3.7 | 7.8±0.7 | 1.9±0.1 |  |
| SCR20-OSL-03 | USU-3766 | 1.20 | - | F: 35% | 1.48±0.04 | 66.8±2.7 | 6.0±0.5 | 1.3±0.1 | 0.17±0.02 |
|  |  |  |  | M: 50% | 1.42±0.04 | 62.5±2.5 | 5.5±0.5 | 1.4±0.1 |  |
|  |  |  |  | C: 15% | 2.59±0.06 | 110.0±4.4 | 8.1±0.7 | 1.8±0.1 |  |
| SCR20-OSL-04 | USU-3767 | 1.75 | - | F: 75% | 1.52±0.04 | 68.0±2.7 | 7.1±0.6 | 1.4±0.1 | 0.16±0.02 |
|  |  |  |  | M: 20% | 1.58±0.04 | 68.5±2.7 | 6.4±0.6 | 1.5±0.1 |  |
|  |  |  |  | C: 5% | 2.00±0.05 | 92.2±3.7 | 13.3±1.2 | 1.4±0.1 |  |
| SCR20-OSL-05 | USU-3768 | 3.45 | - | F: 100% | 1.44±0.04 | 62.6±2.5 | 5.9±0.5 | 1.2±0.1 | 0.13±0.01 |
| SCR20-OSL-06 | USU-3769 | 2.25 | - | F: 100% | 1.51±0.04 | 71.1±2.8 | 7.0±0.6 | 1.6±0.1 | 0.15±0.01 |

[1] Assumed 5±2 % for moisture content in dose rate calculation for all samples.

[2] Dose rate subsamples based on grain size: fine-F (<1.7 mm), medium-M (1.7-16 mm), coarse-C (>16 mm), and weighted proportions (%) of subsamples used with chemistry in gamma dose rate calculation. Beta dose rate uses chemistry from fine fraction (<1.7 mm) only when F:>50%.

[3] Radioelemental concentrations determined using ICP-MS and ICP-AES techniques; dose rate is derived from concentrations by conversion factors from Guérin et al. (2011). Grain-size based internal beta dose rate determined assuming 12.5% K and 400ppm Rb using Mejdahl (1979). Alpha contribution to dose rate determined using an efficiency factor, or 'a-value', of 0.09±0.01 after Rees-Jones (1995).

age of 34.4±4.2 ka (SCR20-OSL-04). The upper part of this section is characterized by a ~2.5 m unit containing well-sorted
fine sands and silts sediments interpreted as aeolian deposits (loess). The lowermost loess sample yielded an IRSL of 10.8±1.3
ka (SCR20-OSL-05), while the uppermost resulted in 3.6±0.4 ka (SCR20-OSL-06).

# 5 Discussion

## 5.1 Arroyo Verde moraine chronology – Marine Isotopic Stage 6 (MIS 6)

Boulder samples from the Arroyo Verde II located on the northern margin of the valley provided similar [10]Be ages (Fig. 3K–L),
while the ones on the south differ significantly. For instance, one sample provided a young age of 132.1±6.2 ka (AV-02, Fig.
3J). Given its small size, we suggest that it could have been exhumed through erosion and moraine degradation, hence exposing





the surface to cosmic radiation at a later date. On the other hand, sample AV-01 provided the oldest age (243.1±11.5 ka, Fig. 3I) of the entire dataset, being about 80–110 ka older than the remaining samples. We hypothesize it might record nuclide inheritance from a previous exposure and exclude it from the mean age calculation. Therefore, the mean age of the moraine

results in 153.0 ka±14.7 ka (Fig. S5). We indicate that ages obtained from the Arroyo Verde II moraines correspond to Marine Isotopic Stage 6 (MIS 6: 191 – 130 ka; Lisiecki and Raymo, 2005), indicating that the ice reached one of the narrowest sections of the upper Río Santa Cruz valley during this stage. Lastly, we did not obtain ages for the Arroyo Verde I, and we suggest that this moraine predates the Arroyo Verde II moraines. However, we acknowledge that the limited number of samples challenges our efforts to precisely determine the age of these moraine complexes.

## 5.2 El Tranquilo moraine chronology – Marine Isotopic Stage 3 (MIS 3)

Geomorphological mapping and stratigraphy indicate that the Lago Argentino glacier expanded at least twice during the last glacial cycle, with the El Tranquilo I moraines forming prior to the El Tranquilo II moraines (see section 4.1). Despite our $^{10}$Be ages for each moraine complex are statistically indistinguishable from each other, our dataset indicates that both El Tranquilo I and II moraines were occupied and abandoned during MIS 3 (57 – 29 ka; Lisiecki and Raymo, 2005), and this advance

constitutes the largest extent of the Argentino glacier lobe during the last glacial cycle.

While the El Tranquilo II moraines provided consistent ages, with a mean age 36.6±1.0 ka (no outliers removed), El Tranquilo I moraines show a scattered distribution of ages across the moraine complex. These moraines on the northern laterals are more developed and better preserved compared to the ones on the central portion of the valley. Consequently, we expect boulders from the northern laterals to provide a closer age to moraine deposition compared to boulders located in the central

portion of the valley. For instance, boulders around valley center rarely exceed half a meter in height, implying exhumation that would result in younger ages. Moreover, the hummocky landforms from the central portion of the valley are heavily affected by post-depositional processes such as aeolian erosion/deposition, landsliding, and alluvial-fan development (Fig. S4). Therefore, we exclude ages (Table 2) younger than the mean age of El Tranquilo II moraines to calculate the mean age of the El Tranquilo I, which results in 44.5±8.0 ka.

The IRSL ages are stratigraphically consistent with the MIS 3 glaciation recorded in the cosmogenic nuclides exposure chronology and closely complement the glacial history presented here. Based on stratigraphy and an IRSL age of ∼50 ka retrieved from coarse sands collected from the lower (though not the lowest) section of the river bank, we indicate that this age represents a minimum limiting age on the timing of glacial advance and outwash deposition (Fig. 5B). Therefore, we suggest that fluvial aggradation was underway by ∼50 ka with the El Tranquilo I moraine abandonment occurring at 44.5±8.0 ka

about 10 km westward. On the other hand, IRSL ages from outwash gravels (∼34–29 ka, Table 3) post-date the El Tranquilo II moraines, providing a close age of outwash abandonment, suggesting that glacier recession occurred shortly after ∼36 ka.

Lastly, two additional IRSL ages from loess deposits capping the El Tranquilo II outwash (lower section = 10.8 ka, upper section = 3.6 ka) indicate sustained aeolian activity for the remainder of the Holocene; highlighting prolonged aeolian deposition, as can be widely observed in the landscape.



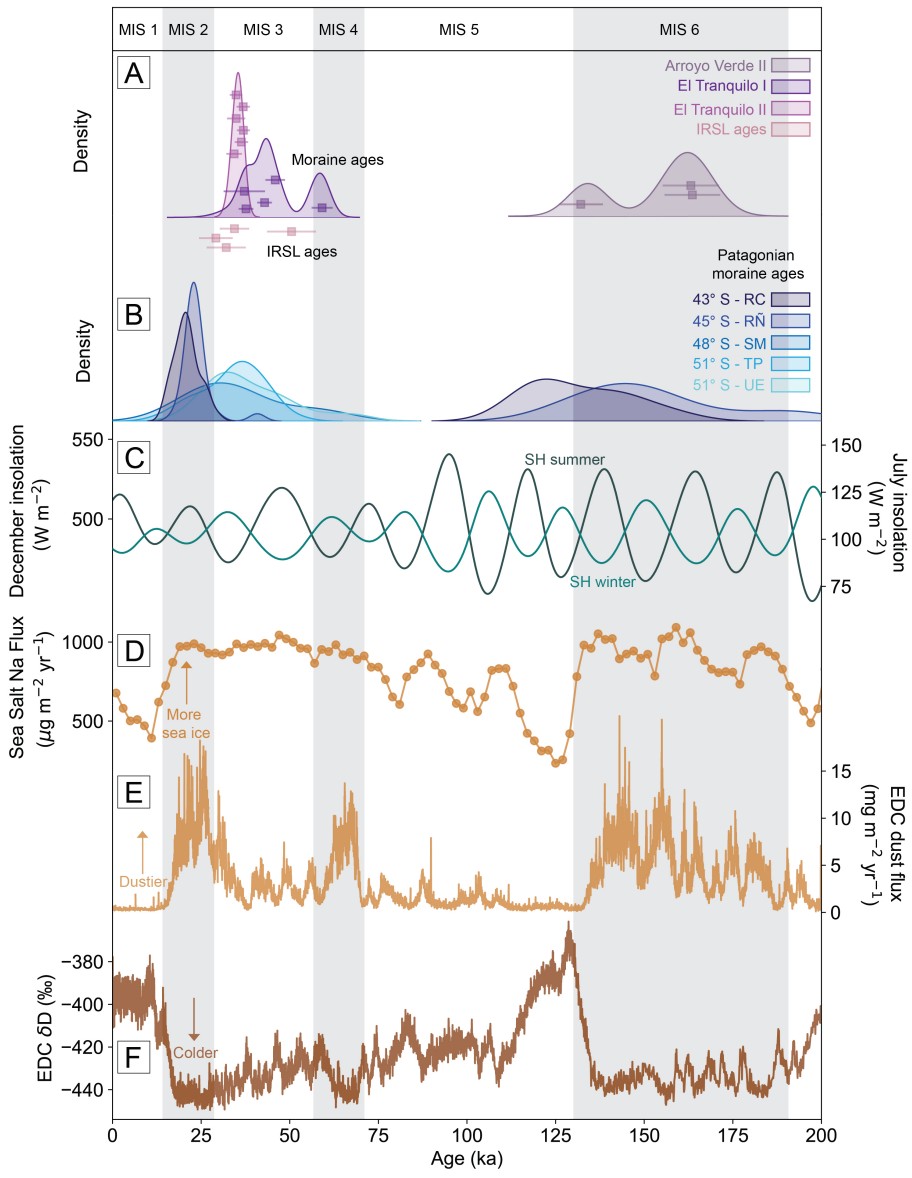

**Figure 6. (A)** [10]Be and IRSL outwash ages for the Lago Argentino outlet lobe, with squares representing the individual ages along with their corresponding error. **(B)** Patagonian moraine age distributions from published studies for the Río Corcovado lobe (RC, Leger et al., 2023), the Río Ñirehuao lobe (RÑ, Peltier et al., 2023), Lago San Martín (SM, Glasser et al., 2011), Torres del Paine (TP, García et al., 2018), and Última Esperanza (UE, García et al., 2018; Sagredo et al., 2011). **(C)** Summer and winter insolation values obtained for Lago Argentino (50° S) for a December average (summer) and July average (winter) after Berger and Loutre (1991). **(D)** Sea Salt Sodium Flux that serves as a proxy for Antarctic sea ice variability from EPICA Dome C ice core (Wolff et al., 2006). **(E)** Dust variability from the EPICA Dome C ice core (Lambert et al., 2008). **(F)** EPICA Dome C $\delta$ deuterium isotope record from the EPICA Dome C ice core (Jouzel et al., 2007) as proxy for temperature variations.



### 5.3 Lago Argentino glacier lobe during Marine Isotopic Stage 6 (MIS 6)

During MIS 6, deuterium isotopes from Antarctic ice cores (Fig. 6F) indicate lower mean atmospheric temperatures (Jouzel et al., 2007) as well as increased dust fluxes inferred to be derived from enhanced rock-flour production (Fig. 6E) (Lambert et al., 2008). Additionally, sea salt sodium fluxes (Fig. 6D), indicative of sustained presence of sea ice, are reported to be greater during glacial periods (Wolff et al., 2006). According to Peltier et al. (2023), glacial expansion in Patagonia would be maximum at both eccentricity and obliquity minima, along with increased sea ice conditions around Antarctica. Our moraine chronology suggests these colder conditions would have promoted glacier expansion at Lago Argentino during MIS 6 at ∼153 ka, highlighting that cold conditions recorded in Antarctic ice cores were also present across Patagonia.

Recently, Leger et al. (2023) indicated that the Río Corcovado outlet lobe, located in northeastern Patagonia, reached its maximum extent during MIS 8, and subsequently advanced during MIS 6 between 150–130 ka. Additionally, Peltier et al. (2023) showed that the MIS 6 ice advance from the Río Ñirehuao system in central Patagonia occurred at ∼153 ka and ∼137 ka. While previous studies (Kaplan et al., 2005; Smedley et al., 2016) focused on glacial deposits east of Lago Buenos Aires (southern Patagonia), evidence of MIS 6 ice extent at this site remains elusive, given that subsequent ice advances could have overprinted older glacial deposits (Hein et al., 2017). Therefore, our study provides one of the first constraints on the timing of the Penultimate Glaciation in southern Patagonia, recorded in the Arroyo Verde II moraines, indicating that glacial conditions were present across northern (Leger et al., 2023), central (Peltier et al., 2023), and southern Patagonia, and was more extensive than the MIS 2 advance. MIS 6 ages have been reported in the southern Alps of New Zealand (Putnam et al., 2013) and southwest Tasmania (Kiernan et al., 2010), suggesting that MIS 6 was more expansive than MIS 2 broadly across the southern mid-to-high latitudes.

### 5.4 Lago Argentino glacier lobe during Marine Isotopic Stage 3 (MIS 3)

Our chronological constraints indicate that the Lago Argentino ice lobe expanded at least twice in upper Río Santa Cruz basin during MIS 3 (Fig. 6A), with fluvial aggradation underway before 50.5±6.9 ka and glacial advance culminating at 44.5±8.0 ka (El Tranquilo I) and at 36.6±1.0 ka (El Tranquilo II). We determine that this outlet lobe reached its maximum extent of the last glacial cycle during MIS 3, when it deposited the El Tranquilo I and II moraines ∼120 km away from the modern ice front.

### 5.4.1 Comparison with other glacial records during Marine Isotopic Stage 3 (MIS 3)

Our new chronology for the Lago Argentino glacier lobe fits well into other evidence showing that some glaciers in the southern mid-latitudes advanced during MIS 4 or 3 and were most extensive than the global LGM (MIS 2). MIS 3 glacial expansions have been widely recognized and dated across Patagonia. Radiocarbon dating on till sequences from the Chilean Lake District (∼42°S), northern Patagonia, determined that glacial expansions occurred at ∼33.6, 30.8 ka and during early MIS 2 in (Denton et al., 1999; Moreno et al., 2015). In central Patagonia, exposure ages indicated that the PIS expanded at Lago San Martín (Fig. 1B) during late MIS 3, culminating between 38–34 ka (Glasser et al., 2011), while Hein et al. (2010) dated the outermost moraine of Lago Pueyrredón (Fig. 1B), providing ages ranging between ∼32–25 ka. Although moraine





data from Lago Buenos Aires (Fig. 1B) resulted in MIS 2 ages (Douglass et al., 2006; Kaplan et al., 2004, 2005), outwash deposits dated by luminescence (Smedley et al., 2016) suggest that the proglacial plains were formed during MIS 3, between ∼34 and 30.8 ka. In southern Patagonia, results from moraine boulders indicate that the Torres del Paine and Última Esperanza

(Fig. 1B) outlet lobes expanded during MIS 3, with ice advances culminating at ∼48, 39, 35 ka (Çiner et al., 2022; García et al., 2018; Girault et al., 2022; Sagredo et al., 2011), indicating that the MIS 2 advance was half the extent of the local LGM during MIS 3 (Fig. 6B). At Seno Skyring (Fig. 1B), Lira et al. (2022) reported full glacial conditions during MIS 2, though implying that some of the samples could reflect previous exposure from earlier glaciations, in concordance with nearby records.

In southernmost South America, Darvill et al. (2015) demonstrated that the Bahia Inútil – San Sebastián (Fig. 1B) glacier

lobe advanced during MIS 3, between ∼45.6 and 30.1 ka, while Peltier et al. (2021) reported that the Magallanes lobe advanced during MIS 4, highlighting that these two lobes occupied their most extensive positions prior to the global LGM. Additionally, glacial conditions predating the global LGM have also been identified and dated in other regions of the southern mid-latitudes. For instance, millennial-scale moraine chronology indicates that glaciers in the South Atlantic region (Mt. Usborne) expanded from ∼45 to 20 ka (Hall et al., 2020). Several lines of evidence support that glaciers in the southern Alps of New Zealand

expanded prior to the global LGM, with the most extensive advance of the last glacial cycle occurring during MIS 4 (Schaefer et al., 2015) or MIS 3 (Putnam et al., 2013; Strand et al., 2019; Shulmeister et al., 2019, 2018; Kelley et al., 2014; Doughty et al., 2015).

### 5.4.2 Comparison with paleoclimate records and possible drivers of glacier fluctuations

Previous studies have investigated the role of insolation in driving a pre-LGM glacier growth in the southern mid-latitudes

(Fig. 6C). Huybers and Denton (2008) hypothesized that Antarctic temperatures were influenced by the duration of the seasons in the Southern Hemisphere. Winter duration was enhanced during MIS 3 relative to MIS 2, which would ultimately promote lower temperatures that were capable of driving glacier growth (Darvill et al., 2016). Additionally, southern winter insolation is thought to have played a role in driving an early MIS 3 glacier advance in Southern Patagonia, since winter insolation was at its minimum during this time ∼48 ka, exacerbating colder winters (García et al., 2018). In contrast, lower summer

insolation intensity towards late MIS 3 at ∼35 ka, along with longer winters, would have favored ice growth during this time, coincident with some of the glacial advances in Patagonia and New Zealand, including our Lago Argentino record presented here. However, these advances culminated at different times and not necessarily in a synchronous fashion, implying that insolation might not be the single forcing responsible for pre-LGM glacier growth across the southern mid-latitudes (García et al., 2018; Darvill et al., 2015, 2016; Doughty et al., 2015; Putnam et al., 2013).

Our Lago Argentino chronology suggests that ice advances culminated broadly in phase with Antarctic stadials, recorded by lower deuterium values and higher dust fluxes at the millennial-scale during MIS 3 in the ice core record (Fig. 6E, F). A progressive trend towards lower temperatures throughout MIS 3, along with distinct millennial-scale variations derived from the atmosphere–Southern-Ocean coupled system in which $CO_2$ exchange promotes seasonal upwelling, is thought to have enhanced millennial-scale $CO_2$ pulses (Gottschalk et al., 2015, 2020). While our record and others (Sagredo et al., 2011; García

et al., 2018; Kelley et al., 2014; Strand et al., 2019; Putnam et al., 2013; Denton et al., 1999) suggest synchronous cooling in



Antarctica and the southern mid-latitudes, our data compilation indicates that these in-phase changes may not have been ubiquitous (Darvill et al., 2016). This could be explained by the fact that lower atmospheric temperatures during MIS 3 would also have enhanced sea-ice formation and expansion around Antarctica (Fig. 6D) (Fogwill et al., 2015; Sigman et al., 2004; Wolff et al., 2006) as well as ocean stratification, as evidenced in lower opal fluxes in the Southern Ocean indicative of decreased

upwelling (Anderson et al., 2009). In turn, these conditions would have promoted a northward shift of the Subantarctic front, which is thought to be responsible for driving both strengthening and an equatorward shift of the core of SWW belt (Kohfeld et al., 2013). This migration would have delivered increased precipitation over some zones of the southern mid-latitudes, modulating glacier mass balance at the millennial-scale, allowing glaciers in the southern latitudes to advance earlier in the glacial cycle, such as during MIS 3 (Darvill et al., 2015, 2016; Hall et al., 2020; Shulmeister et al., 2019).

These conditions driven by feedbacks between the atmosphere–Southern-Ocean coupled system, modulated by orbital parameters, would have resulted in pre-LGM glacial advances at Lago Argentino during MIS 3. We suggest that sustained lower atmospheric temperatures by late MIS 3 towards MIS 2, as well as a northward migration of the SWW belt, would have promoted widespread ice growth across Patagonia, coincident with some other Patagonian lobes that experienced their local glacial maximum at ∼35 ka (Davies et al., 2020). This equatorial migration would have decreased precipitation over the southern lat-

itudes during MIS 2, reducing ice extent compared to MIS 3 (Fogwill et al., 2015; Hall et al., 2020), which could be partly due to a westward migration of the ice divide (García et al., 2018; Sugden et al., 2002). This implies that moisture-starved glaciers would have not reached their prior extent, highlighting the role of increased precipitation in driving an early glacial maximum across the southern mid-latitudes (Darvill et al., 2015, 2016; Rother et al., 2014; García et al., 2018; Shulmeister et al., 2019). Additionally, we hypothesize that increased precipitation during MIS 3 would have been favorable for glaciers to develop a

temperate regime, as indicated for some of the southernmost outlet lobes of the PIS (Darvill et al., 2017). At eastern Lago Argentino, this is evidenced in lateral meltwater channels as well as widespread presence of drumlins and lineations inboard the moraine complexes. In turn, lower shear stresses at the glacier base favored by this thermal regime would have promoted enhanced basal sliding, allowing for glaciers to extend further to the east.

## 5.5    Landscape evolution synthesis

The Arroyo Verde I moraines presumably indicate the easternmost glacier extent of the Lago Argentino outlet lobe at the upper basin of the Río Santa Cruz during the mid-Pleistocene, where the valley reaches its narrowest point (Fig. 7A). Additional stratigraphic and glaciotectonic data indicate that the extension of this advance was topographically controlled and limited by the existing Pliocene plateaus located at the northern and southern margins of the valley (Goyanes and Massabie, 2015), as well as preceding west-to-east faulting (Glasser and Ghiglione, 2009). Geomorphological and chronological evidence presented

here is therefore indicative that the outlet lobe did not reach elevations above the plateaus (∼800–1000 m) during the Arroyo Verde glacial advances.

Glacial deposits from this advance are found in the vicinity of Río La Leona. The asymmetric nature of the morphology of the valley, which is deeper at the northern margin, as well as the orientation of the subglacial landforms, suggests that the Lago



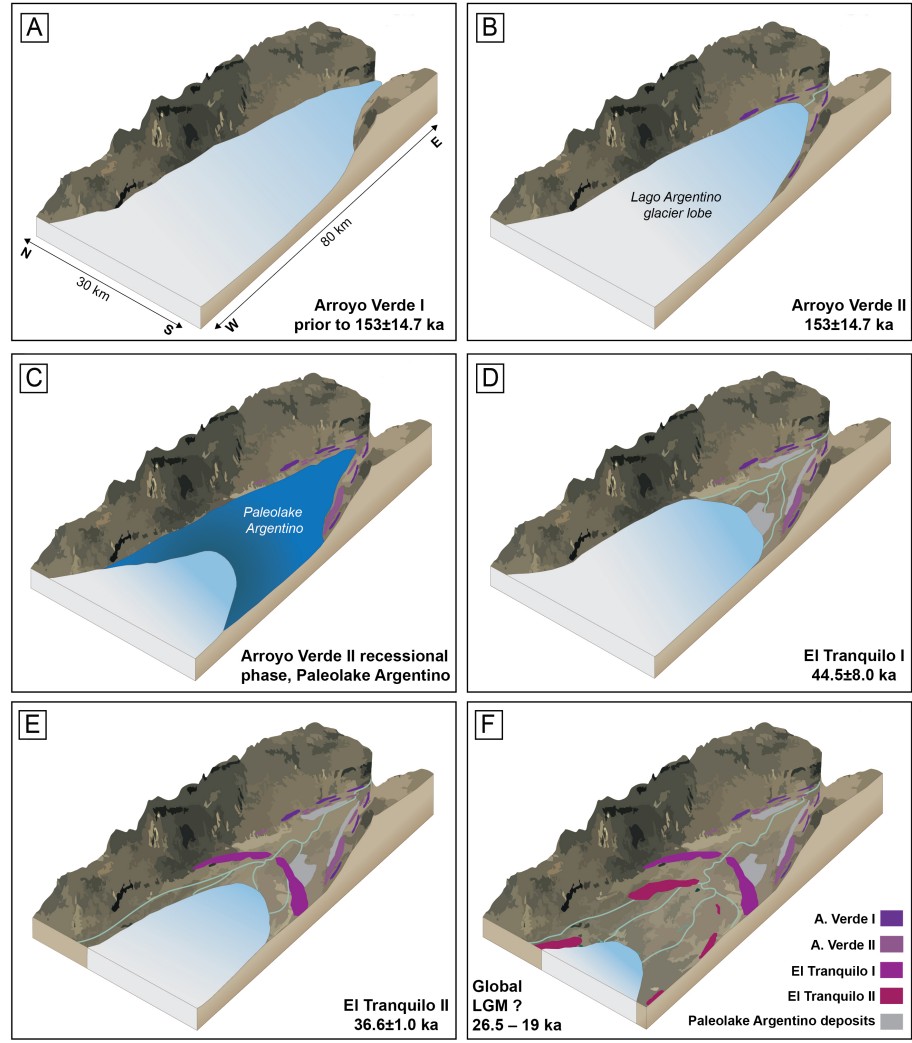

**Figure 7.** Landscape evolution conceptual model developed for the upper basin of the Río Santa Cruz and eastern margin of Lago Argentino, based on geochronological and geomorphological data presented here along with published interpretations (Strelin and Malagnino, 1996; Strelin et al., 1999; Strelin and Malagnino, 2009). **(A)** Easternmost extent of the Lago Argentino outlet lobe during the Arroyo Verde I phase prior to ~153 ka, **(B)** extent during the Arroyo Verde II phase at ~153 ka, **(C)** Arroyo Verde II recessional phase along with development of an extensive paleolake (paleolake Argentino) likely during the Penultimate Deglaciation, **(C)** Readvance of the Lago Argentino glacier with moraine abandonment occurring at ~44.5 ka (El Tranquilo I moraines), **(E)** Readvance of the Lago Argentino glacier at ~36.5 ka (El Tranquilo I moraines), and **(F)** Recession after ~36.5 ka, with ice present at the margin of the lake, presumably during the global LGM (26 – 19 ka).





Argentino lobe merged with tributary glaciers from the Guanaco, Turbio, and Viedma basins during this interval (Strelin and
Malagnino, 1996).

Strelin and Malagnino (1996) suggested that widespread glacier recession took place after the deposition of the Arroyo Verde
I moraine, and pointed out that the geomorphic evidence is not conclusive of whether the following advance (i.e., Arroyo Verde
II) is part of the same glaciation or not. Our chronological constraints highlight an avenue for future research, since a single
sample yielded an MIS 8 age, that could indicate that the Arroyo Verde I moraines could have been deposited during this
glacial period. On the other hand, results from cosmogenic nuclide surface exposure dating presented here indicate that the
Lago Argentino lobe advanced during MIS 6, culminating at 153±14.7 ka, depositing the Arroyo Verde II moraines.

Later, a large proglacial lake developed, reaching elevations of ∼280 masl, previously recognized as paleolake Argentino
(Strelin and Malagnino, 1996) and is represented by the glaciolacustrine sediments mapped in Fig. 2. Widespread lacustrine
landforms, particularly at the southern margin of the valley, suggests that this paleolake lasted long enough in order to develop
deltas, spits, and shorelines. Strelin and Malagnino (1996) suggested that this paleolake formed during the Arroyo Verde
II recessional phase in response to valley over-deepening (Strelin and Malagnino, 1996, 2009), where the Arroyo Verde II
moraines served as a dam (Fig. 7C). Afterward, fluvial routing eroded these moraines, causing the paleolake to eventually
drain. This paleolake is then thought to have been drained abruptly in response to moraine degradation by fluvial erosion
during its latest stages, as evidenced in mega-ripples located downstream (Strelin and Malagnino, 1996, 2009).

Fluvial aggradation was underway by ∼50.5±6.9 ka, indicating a minimum-limiting age for the following ice advance
during MIS 3, while the Lago Argentino glacier lobe abandoned the El Tranquilo I moraines at ∼44.5±8.0 ka (Fig. 7D).
Moraine deposits from this stage were correlated with glacial deposits at higher elevations at the northern margin of the valley,
suggesting that the Lago Argentino outlet lobe reached the southern part of Río La Leona valley (Strelin and Malagnino,
1996). Additionally, fault planes within deformed glaciolacustrine sediments evidence a stress transfer from the north side
to the southeast, as well as a west-to-east pushing due to glacier loading (Goyanes and Massabie, 2015). The Lago Argentino
glacier developed an extensive glaciofluvial plain during this stage that was later partially eroded by the El Tranquilo II outwash,
resulting in the incision of the previously deposited outwash plain and filling of younger outwash (cut-and-fill).

Inboard of the El Tranquilo I moraines, the Tranquilo II moraines were deposited at ∼36.6±1.0 ka during a less extensive
ice advance that did not reach the Río La Leona valley, being spatially restricted to the northeastern margin of Lago Argentino
(Fig. 7E). Here, paleochannels reveal that the Río La Leona drained into Río Santa Cruz as the ice occupied the eastern margins
of Lago Argentino. Later, the Río La Leona changed its drainage route by draining into the lake after the ice lobe abandoned
the margins of Lago Argentino (Fig. 7E) and receded westward (Strelin and Malagnino, 1996). Strelin and Malagnino (1996)
indicated that during this time the lake level dropped from about 230 to 210 meters as the moraine circling Lago Argentino was
subsequently eroded. While evidence of a MIS 2 advance is lacking in our terrestrial geomorphic record, we speculate that it
could be deposited within the lake basin or covered by the extensive aeolian deposits located at the eastern margin of the lake
(Fig. 7F). Later, widespread recession took place until the following advance occurred during the Late Glacial. This advance
is recorded in the Puerto Banderas moraines, about 45 km west of the margins of Lago Argentino (Strelin et al., 2011).



Finally, we indicate that our IRSL ages document the onset of loess deposition at ∼10.8 ka in southern Patagonia, as well as sustained aeolian activity during the Holocene. Our ages (10.8 and 3.6 ka) coincide with warm and dry intervals recorded

by changes in pollen assemblages retrieved from Lago Cipreses (Moreno et al., 2018), which is located less than 100 km south of Lago Argentino. Additionally, no evidence for glacial advances is available from southern Patagonia during these periods (Strelin et al., 2014; Kaplan et al., 2016; Reynhout et al., 2019).

## 6 Conclusions

We investigate the glaciogenic landforms at the upper basin of Río Santa Cruz, eastern Lago Argentino, and we identify at least

420 four moraine complexes (i.e., Arroyo Verde I and II; El Tranquilo I and II) through high-resolution geomorphological mapping. We use two independent geochronological techniques to provide the first published constraints on the timing of the middle-to-late Pleistocene glaciations by employing [10]Be cosmogenic nuclide surface exposure dating on boulders and IRSL on outwash sediments. We determine that the Lago Argentino glacier, an outlet lobe of the former Patagonian Ice Sheet, expanded at 153.0±14.7 ka during Marine Isotope Stage 6 (the Penultimate Glaciation) in the upper basin of the Río Santa Cruz, where it

deposited the Arroyo Verde II moraines. Additionally, we find that fluvial aggradation due to glacier advance was underway by 50.5±6.9 ka and that El Tranquilo I moraines were deposited during Marine Isotope Stage 3 at 44.5±8.0 ka. This advance precedes the global Last Glacial Maximum, and constitutes the most extensive advance of the Lago Argentino glacier during the last glacial cycle, being consistent with published records around the southern mid-latitudes. We determine that the second pulse of this advance (i.e., El Tranquilo II) culminated at 36.6±1.0 ka, while proglacial plains became abandoned between

34–29 ka. We hypothesize that during MIS 2 the glacier did not occupy the Río Santa Cruz upper basin and likely remained limited to the margins of the modern Lago Argentino, highlighting an avenue for future research.

Based on our new chronology, we find that glacial conditions recorded in Antarctic ice cores were also present and widespread across Patagonia, both in the north and in the south, during the MIS 6 Penultimate Glaciation. We hypothesize that the duration of the Southern Hemisphere winters along with minimum summer insolation promoted colder conditions during MIS 3.

Ultimately, that derived in an equatorial shift of the Southern Westerly Winds (SWW) belt, resulting in enhanced precipitation in the southern latitudes, that drove, in turn, glacial expansion at Lago Argentino. Our record coincides with evidence from different glacier lobes nearby (e.g., Torres del Paine, Última Esperanza, and San Sebastián – Bahia Inutil) and highlights the role of millennial-scale fluctuations of the latitudinal position of the SWW belt in modulating the mass balance of mid-latitude glaciers during the last glacial cycle. We suggest that increased precipitation would have induced a temperate regime of the

Lago Argentino glacier, favoring basal sliding and causing the glacier to reach its most extensive position during the last glacial cycle before MIS 2. Lastly, we document the onset of aeolian deposition after ∼10.8 ka following warmer and drier conditions in southern Patagonia, as evidenced is pollen records nearby.





*Data availability.* All data associated with the production of new [10]Be exposure ages and IRSL ages, including field and analytical data, are provided in the manuscript tables and figures and in the supporting documents in the supplementary material. The 5 m digital elevation
model is available from Instituto Geográfico Nacional at https://www.ign.gob.ar/NuestrasActividades/Geodesia/ModeloDigitalElevaciones/ Introduccion. Global bathymetry data is available from the GEBCO website at https://www.gebco.net/data_and_products/gridded_bathymetry_ data. Wind speed data available from the NCEP-NCAR Reanalysis 1 at https://psl.noaa.gov/data/gridded/data.ncep.reanalysis.html. Outlines (ice sheet extent, rivers, lakes, and icefield) used in for Fig. 1B were adapted from the PATICE project (Davies et al., 2020). Insolation curves obtained for Lago Argentino area were derived using climlab (https://climlab.readthedocs.io/en/latest/index.html).

*Author contributions.* MR, SBP, MV, and ADW conceived the study. MR, AGJ, and SAM carried out [10]Be extraction. SBP and TR analyzed IRSL samples. MR generated maps with inputs from MAM and JAS. MR processed the data, and wrote the manuscript with input from all authors.

*Competing interests.* The authors declare no competing interests.

*Acknowledgements.* The authors would like to acknowledge to the owners of the sites visited who granted access. We thank Anastasia
Fedotova and Guillermo Tamburini-Beliveau for field support, as well locals in El Calafate who provided directions and facilitated the expedition. Rock crushing was carried out at the STAMP Lab at the University of Minnesota-Twin Cities, therefore we thank the research staff who assisted in the process. The authors thank Dr. Michael R. Kaplan for early conversations that greatly improved the interpretation of the dataset, as well as Dr. Phillip Larson and Dr. Victoria M. Fernandes for the insights on fluvial geomorphology. Additionally, we thank Dr. Eduardo Malagnino for feedback on glacial geomorphological mapping. We also thank the colleagues from the GUANACO project for
conversations that inspired and improved this work. This material is based upon work supported by the National Science Foundation under a collaborative research award Grant No. EAR-1714614 to Wickert, Ito, and Noren, coordinated by Lead PI Maria Beatrice Magnani.



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
