# Peer review of "Late Quaternary glacial maxima in Southern Patagonia: insights from the Lago Argentino glacier lobe"

_Climate of the Past, 2024_

## Author Comment (AC1)

**Late Quaternary glacial maxima in Southern Patagonia: insights from the Lago Argentino glacier lobe**

**Romero et al. (2024)**

We thank both reviewers for their positive and thoughtful commentary on the manuscript. The proposed changes undoubtedly improve this work. The two reviewers shared similar feedback about the interpretation of the older landforms (Arroyo Verde moraines), so we provide a single response for addressing this concern. Minor comments are addressed individually. The attached document contains the reply to the reviewer.

Reviewer comments are presented in their original format and our response is in blue text, with manuscript changes in blue and **bold**. To streamline review, for small changes on grammar, word choice, etc. we write 'Done', 'Addressed', and/or 'Noted' to signify we have made the change.

**Authors reply to Dr. Andrew Hein – RC1**

This is an interesting and valuable manuscript that introduces new geomorphological mapping and geochronology (cosmogenic [10]Be and IRSL dating) that firmly establishes the timing of the local Last Glacial Maximum (LGM) of the Lago Argentino glacier lobe. This glacier lobe reached its maximum during Marine Isotope Stage (MIS) 3, at about 45 ka and again at about 37 ka, rather than during the global LGM, which occurred more recently during MIS 2. It also shows that there are older glacial moraines preserved in the valley that could indicate a more extensive advance dating to the penultimate MIS 6 glaciation (~160 ka), but that is less certain as I will discuss later. The authors find no evidence for glacial advances in the valley that correspond in timing to the global LGM and suggest such deposits could be less extensive and therefore remain preserved beneath the present lake. The overall timing of the local maximum is quite similar to other valleys in southern Patagonia, such as the Torres del Paine and Ultima Esperanza valleys just to the south. The authors suggest ideas as to why the local LGM occurred earlier than the global LGM, inferring more intense winters and a northward expansion of the southern westerly winds (SWW) causing an increase in precipitation during MIS 3.

The manuscript makes a valuable contribution to knowledge on glacier behaviour during the last glacial cycle in a part of Patagonia where this data is currently limited. The paper's thrust tackles the local LGM, and the data presented supports the interpretations made. However, the interpretations made regarding the age of the older moraine system are, in my view, not supported by the data. I would suggest some changes to this part of the manuscript before publication. When those changes are made, I would recommend publication of the manuscript.

Dr. Hein's points about the older moraine system are well-taken, and we modify the interpretation and discussion throughout the manuscript, which we detail below.

Main Revisions

The cosmogenic [10]Be ages from moraine boulders on the Arroyo Verde II moraines are few and widely scattered. Four exposure ages range from 132-243 ka. While two of the exposure ages are similar (~160 ka), the data are insufficient to conclude that this moraine was formed during MIS 6. It may well prove to be a correct age assignment, but the current data are not sufficient evidence. I recommend the manuscript is changed to indicate that this age assignment is no more than tentative, given the paucity of data to support the age. This will require changes in the abstract, conclusions and throughout the body

of the text (e.g., section 5.3) and figures (e.g., Figure 6) where the MIS 6 age is currently indicated as the definitive age of the advance.

There has been a significant amount of published work in similarly dry environments in neighbouring valleys within Argentine Patagonia demonstrating that outwash terrace surfaces are better targets for exposure dating of "old" glacial advances (i.e., pre-last glacial cycle) than corresponding moraine limits, which tend have scattered and too-young exposure ages (e.g., Hein et al., 2009; 2011; 2017; Darvill et al., 2015; Mendelova et al., 2020; Leger et al., 2023).  In the Lago Pueyrredón and Lago Buenos Aires valleys', the outwash terraces linked directly to (and in between) moraines with MIS 6 boulder ages, instead had MIS 8 outwash cobble ages (i.e., 100 ka older), suggesting moraine degradation had led to erroneously young boulder exposure ages (Hein et al., 2009; 2017).  In a more extreme case, moraine boulders in the Lago Pueyrredón valley with exposure ages mostly between 190 – 230 ka, but with a single 600 ka outlier, had corresponding outwash ages of ~600 ka, consistent with the age of the oldest boulder outlier (suggesting all of the other moraine boulders were too young). Given this regional context from neighbouring valleys, it could be the case that the 243 ka boulder age in this study is closer to the age of the Arroyo Verde II moraine.  In other words, it may very well be that the Arroyo Verde II moraine was deposited during MIS 8 instead of MIS 6, much like the "Hatcher" and "Moreno" moraine systems in the Lago Pueyrredón and Lago Buenos Aires valleys a little further north (47S) in central Patagonia. The revisions to the present manuscript should leave open the possibility that the sampled moraines could indeed be older.  Moving forward, this could be tested by increasing the dataset and targeting the outwash cobbles. If both outwash and moraine boulders give MIS 6 ages, then the uncertainty in age assignment would be greatly reduced (e.g., Leger et al., 2023).

We appreciate both reviewers' thorough commentary on this section of the manuscript. We concur with their main concern about our interpretation of the older glacial landforms mapped and dated in this work. Accordingly, we provide a modified version of our interpretations to account for the fact that the nature of our dataset can not precisely resolve the timing of occurrence of these landforms to an individual Marine Isotope Stage. Therefore, we acknowledge that given our small number of samples and the existing literature on glacial landforms older than 100 ka (Hein et al., 2009; 2011; 2017; Darvill et al., 2015; Mendelova et al., 2020; Leger et al., 2023), the Arroyo Verde moraines could correspond to MIS 6 or MIS 8. This extends our interpretation to include an older advance and highlights an avenue for future research. This change is reflected throughout the paper, including in the abstract, conclusion, and Figure 6.

Suggested minor changes

Abstract:L0-5: "Despite synchronous ice-volume and extent change across hemispheres, evidence from the southern mid-latitudes indicates that local glacial maxima occurred earlier in the glacial cycle" – this sentence appears to contradict itself.  Suggest re-writing.

We rewrite the second sentence of the abstract as follows: **Determining the timing and extent of Quaternary glaciations around the globe is critical to understanding the drivers behind climate change and glacier fluctuations. Evidence from the southern mid-latitudes indicates that the local glacial maximum preceded the global Last Glacial Maximum (LGM), implying that feedbacks in the climate system or ice dynamics played a role beyond the underlying orbital forcings.**

L35: It's not necessarily that they are poorly preserved to the west, but the LGM ice sheet probably terminated in the sea, so fewer moraines exist to the west.

We agree with this comment and re-arranged the sentence accordingly: **Lago Argentino drains to the east into the Río Santa Cruz basin and, ultimately, the Atlantic Ocean. Given that during the LGM the western margin of the PIS reached the sea (Davies et al., 2020), fewer moraines are preserved in the terrestrial geological record. In contrast, glacial landforms in the arid eastern foreland of the Andes are better preserved due to lower weathering rates related to the rain shadow effect imposed by Andean Range (Garreaud, 2009), making moraines located in the Argentine steppe more suitable for geochronological dating.**

L40: Capitalise "Late" Glacial

**Done.**

L115: I have sampled outwash cobbles in the Lago Argentino valley, and looked at several boulders. Many of these were fluted and ventifacted as a consequence of being downwind of the dust source (outwash plains). I would imagine this type of aeolian erosion would be most severe in the centre of the valley. Was this observed? Did the samples collected have evidence for aeolian erosion?

We observed widespread aeolian deposits in the center of the valley, mainly parallel to the Rio Santa Cruz, and we decided not to sample partially buried boulders or those that exhibited strong signs of aeolian erosion. These aeolian deposits cover a substantial portion of the center of the valley, mainly where the El Tranquilo II limit is hypothesized to be.

Please indicate whether such aeolian erosion is present (and its severity) at an appropriate point in the manuscript.

We do note (L264) that some of our younger ages obtained from boulders from the El Tranquilo I advance in the center of the valley could be partly due to more severe aeolian erosion, amongst other processes such as landsliding and alluvial fan action.

Please also note any evidence for moraine degradation at the sample sites (I note from Figure 3I that boulder AV-01 appears to have ~50 cm of recent exhumation). Is this common across the sampling sites?

The most evident sign of moraine degradation, and therefore, boulder exhumation, was found in the site where we sampled boulder AV-01. As noted by the reviewer, this boulder exhibits about half a meter of exhumation. However, we did not observe boulders with significant signs of such exhumation as boulder AV-01. We now note that the other boulders from the Arroyo Verde moraines could have experienced exhumation, and we adapt our interpretations accordingly (see major revisions).

L131: I would encourage aborting use the 1.4 mm ka-1 erosion rate, but instead use the 0.2 mm ka-1 rate from Douglass et al., 2007. Kaplan always meant for the 1.4 rate to indicate a "maximum" erosion rate, but it seems to have been adopted as "the" erosion rate in a lot of literature – check discussion on this topic in Hein et al. (2017). If it is used, please explicitly indicate this is a maximum erosion rate.

We now describe the Kaplan erosion rate as a maximum bound and modify the text accordingly: **As part of a sensitivity test, we calculated the ages with different erosion rates for all landforms, ranging from**

**0.2–1.4 mm ka⁻¹ according to Douglass et al. (2006) and Kaplan et al. (2005), respectively, with the latter representing a maximum erosion rate.**

L175: state how far west of Arroyo Verde I in km (for consistency with subsequent description)

We state that the frontal margin of the Arroyo Verde II moraines is located about 10 km west of the Arroyo Verde I moraines and modify the text as follows: **The frontal margin of the Arroyo Verde II moraines is located ~10 km west of the Arroyo Verde I moraines and comprise multiple ridges and low-relief hummocks (~10 m).**

L175: It is not clear from where the elevations of the moraines are taken from, the terminal moraine elevation? The maximum height in the mapping area east of the lake? Or is it just the maximum elevation before they are no longer traceable?

Moraine hummock elevations were obtained by measuring the difference between the hummock crest and the surrounding flat terrain, in order to capture the relief of the landform. This was performed in several hummocks across the landscape, both frontal and lateral, and we report the average of those relief measurements. That elevation does not reflect the elevation above sea level, or a maximum elevation, but rather represents an average local relief.

L194: remove "in" before "Strelin and Malagnino"

Done.

Figure 2: hard to see palaeo channels

We modified the figure to make the paleochannels more noticeable.

Section 5.1: Earlier the young age of 132 ka was rejected as recently exposed through exhumation, so I'm wondering why it is included in the calculation of the mean age of 153 ka. Is a mean age sensible with so few data, and such large geological scatter? Some take the oldest boulder age as a strategy (e.g., Kaplan et al., 2005). Please explain interpretation here.

We revisited our interpretation of the Arroyo Verde moraines as noted earlier, and we now include all the ages for our interpretations.

L243: "Isotope" rather than "Isotopic"

Done.

L255: "Isotope" rather than "Isotopic"

Done.

Section 5.2: a potential added issue for the samples from the centre of the valley is subsequent fluvial erosion causing undercutting/slumping of moraines and therefore degradation/exhumation of boulders if the sampled surfaces were in contact with former drainage routes.

We agree and expand our rationale for being cautious of boulders located in the center of the valley, especially when it comes to proglacial drainage. L265 is modified as follows: **Consequently, we expect boulders from the northern lateral moraines to provide a closer age to moraine deposition compared**

**to boulders located in the central portion of the valley. For instance, boulders around the valley center rarely exceed half a meter in height, making these smaller boulders more susceptible to exhumation, which could result in younger ages. The hummocky landforms from the central portion of the valley are heavily affected by post-depositional processes such as aeolian erosion/deposition, landsliding, and alluvial-fan development (Fig. S4). Additionally, samples from landforms located in the valley center could have been affected by former proglacial river-discharge routes, as noted in paleochannels described in section 4.1.4.**

Figure 6: Depending on how the MIS 6/8 discussion evolves, it may be sensible to increase the age scale to include MIS 8 and the chronologies that align with that age.

We modified the figure to accommodate the chronologies that align with MIS 8.

Please indicate how the "Patagonian moraine age distribution" sites were chosen for this figure. For example, other major valleys not mentioned include Lago Buenos Aires, Lago Pueyrredón, and others.

The studies behind the Patagonian moraine age distribution plot were selected because of their proximity to Lago Argentino (San Martin, Ultima Esperanza, Torres del Paine) and/or because they provided both ages for MIS 6 (Rio Corcovado, Rio Ñirehuao). For the sake of expanding the discussion, we include additional relevant sites to this plot, such as Lago Buenos Aires and Lago Pueyrredón to include the work done on the MIS 6 and MIS 8 glacials.

Section 5.3: A relevant reference may be a new marine core off the Chilean coast that indicates glacier activity during MIS 6. Hagemann et al., 2024: https://doi.org/10.1073/pnas.230298312

Noted and added this reference to the discussion.

L300: I don't think there is evidence that these are separate re-advances? If so, make this clear (i.e., cross-cutting relationships, etc).

We stress that the Tranquilo I and II moraines were two separate advances of the Lago Argentino glacier lobe during MIS 3. For instance, we note that the Tranquilo II moraines are located at least ~10 km west of the El Tranquilo I moraines (L182). These advances are recognized by their associated proglacial outwash plain outboard of the moraine complexes and their drumlinized terrain inboard of them. Notably, the El Tranquilo II outwash is found within valleys incised into the El Tranquilo I outwash, serving as a cross-cutting relationship (L200, L403). This is also evidenced in our stratigraphic logging and IRSL ages that show that glaciofluvial aggradation was underway by ~50 ka, with cut-and-fill outwash deposition occurring at ~32 ka, postdating the El Tranquilo II advance. Although various sections of the earlier manuscript did describe the lines of evidence behind our reasoning for this argument, we modified section 5.4 to make it more clear: **Our chronological and geomorphological constraints indicate that the Lago Argentino ice lobe expanded at least twice into the upper Río Santa Cruz basin during MIS 3 (Fig.6A), with fluvial aggradation underway before 50.5±6.9 ka and glacial advances culminating at 44.5±8.0 ka (El Tranquilo I) and at 36.6±1.0 ka (El Tranquilo II). We determine that this outlet lobe reached its maximum extent of the last glacial cycle during MIS 3, when it deposited the El Tranquilo I and II moraines ~120 km away from the modern ice front. Cross-cutting relationships informed by El Tranquilo II outwash deposited within valleys incised into the El Tranquilo I outwash allow us to determine that these were two different advances of the Lago Argentino glacier lobe during MIS 3.**

L306: "and were more extensive than during the global LGM"

Noted and modified.

L315: The reference to çiner et al. 2022 should be under "c" in the reference list

Done.

L350 – Currently the core of westerlies is at about 50 S (i.e., Lago Argentino), so it seems cooling would be more important than a northward migration of the westerlies?

We note cooling as the main driver of glacier change, while millennial-scale changes in the position of the westerlies belt would modulate the timing of glacial advance, as invoked for different sites in Southern Patagonia (Sagredo et al., 2011; Darvill et al., 2015; 2016; Garcia et al., 2018; Hall et al., 2020) and New Zealand (Shulmeister et al., 2019; Strand et al., 2019), highlighting the role of precipitation in influencing the mass balance of some glaciers lobes of the former PIS. New studies off the coast of Chile, as pointed by the reviewer, indicate that MIS 4 advances in Northern Patagonia could have been caused by cooling and precipitation delivered by a northward shift of the westerlies (Spronson et al., 2024; Hagemann et al. 2024), while the core of the westerlies could have been extended over Lago Argentino during MIS 3 at a glacial erosion hotspot defined by Herman and Brandon et al. (2015). We therefore make changes to section 5.4.2 to make a more consistent argument indicating that underlying orbital parameters behind cooling (longer and colder winters) and additional feedbacks controlled by the Southern Ocean–atmosphere coupled system displaced and strengthened the westerly belt over southern Patagonia, particularly at ~50 S (Lago Argentino), delivering increased precipitation that caused glacier lobes to reach their most extensive preserved advance during MIS 3.

L361 – suggest adding Mendelova 2020 to that reference list

Done.

---

## Author Comment (AC2)

**Late Quaternary glacial maxima in Southern Patagonia: insights from the Lago Argentino glacier lobe**

**Romero et al. (2024)**

We thank both reviewers for their positive and thoughtful commentary on the manuscript. The proposed changes undoubtedly improve this work. The two reviewers shared similar feedback about the interpretation of the older landforms (Arroyo Verde moraines), so we provide a single response for addressing this concern. Minor comments are addressed individually. The attached document contains the reply to the reviewer.

Reviewer comments are presented in their original format and our response is in blue text, with manuscript changes in blue and **bold**. To streamline review, for small changes on grammar, word choice, etc. we write 'Done', 'Addressed', and/or 'Noted' to signify we have made the change.

**Authors reply to Dr. Christopher Darvill – RC2**

This manuscript presents an important new reconstruction of the Lago Argentino glacier in Argentine Patagonia based on geomorphological mapping and Be-10 and IRSL dating of glacial landforms. The authors' principle finding is that the glacier reached its maximum extent during Marine Isotope Stages 6 and 3, with no evidence found for an MIS 2 / global Last Glacial Maximum limit (although they note it may still exist). The new evidence adds to a growing body of work from the Southern Andes showing numerous glaciers were more extensive prior to the global Last Glacial Maximum during the last glacial cycle. They add weight to ideas in the literature that this could have been caused by longer, colder winters earlier during the last glacial cycle, combined with movement of the Southern Westerly Wind system. Overall, the paper provides a valuable new record for pre-LGM glacial activity in Patagonia and will be a useful addition to the literature. The paper is well-written and presented, with excellent figures and clear discussion. This is an impressive piece of work. I have only one major issue relating to the authors' interpretation of older ages in the Lago Argentino system—which I believe needs amending throughout the manuscript—and offer an additional list of minor changes that I hope might also strengthen their final paper.

Dr. Darvill's points about the older moraine system are well-taken, and we modify the interpretation and discussion throughout the manuscript, which we detail below.

Major revision

The MIS 3 glacial limit is convincing; the older glacial limit assigned to MIS 6 much less so and should be handled more speculatively. The manuscript currently presents this limit as unambiguously dating to MIS 6 within the abstract, discussion and conclusion, but there are two major issues with this interpretation based on the data presented:

The older Arroyo Verde II limit is assigned to MIS 6 based on only four exposure ages: two in good agreement at ~163 ka, one younger at ~132 (roughly on the MIS 5/6 boundary), and one older at ~243 ka (roughly on the MIS 7/8 boundary). What these clearly show is that the Lago Argentino glacier was more extensive than its MIS 3 El Tranquilo I limit at some point during the Mid-to-Late Quaternary: a valuable conclusion and important addition to the literature. However, the conclusion that this is an unequivocal MIS 6 glacial limit seems too strong, particularly without other supporting evidence (e.g. additional dating approaches and limiting glacial limits, as per Leger et al., 2023, Climate of the Past). I find the rationale for

excluding the older exposure age unclear and not reflective of other approaches in the literature, and either way do not see a strong enough body of evidence for such a strong MIS 6 conclusion. In addition:

Older (e.g. > MIS 2) glacial limits across the Southern Andes have proven challenging to date, with a large body of work demonstrating potential issues, particularly involving dating boulders on moraines like this study. There are lots of examples from the literature, but a good starting point is Hein et al. (2017; Quaternary Science Reviews). Hein et al. show that boulders on older limits may typically underestimate glacial activity quite substantially (e.g. by a glacial cycle). This makes rejection of the older exposure age and categorical assignment of the Arroyo Verde II limit to MIS 6 problematic. Even without the older age, so few ages should be treated more cautiously, particularly given work around Lago Buenos Aires has shown substantial numbers of moraine boulders may still underestimate the timing of glacial retreat.

In summary, it is quite possible the Arroyo Verde II limit does indeed date to MIS 6, but I do not find sufficiently convincing evidence presented here for that conclusion to be so strong. Better to consider MIS 6 as a possibility, but also to consider that the limit may be older. The older exposure age should not be rejected so readily, and the abstract, discussion and conclusion updated to consider a variety of possibilities. I also urge the authors to update their Figure 6, both to factor in the potential for an older limit (i.e. include the older age), and to consider a greater number of existing records in Panel (B) to reflect the fact that numerous MIS 8 limits also exist in Patagonia.

We appreciate both reviewers' thorough commentary on this section of the manuscript. We concur with their main concern about our interpretation of the older glacial landforms mapped and dated in this work. Accordingly, we provide a modified version of our interpretations to account for the fact that the nature of our dataset cannot precisely resolve the timing of occurrence of these landforms to an individual Marine Isotope Stage. Therefore, we acknowledge that given our small number of samples and the existing literature on glacial landforms older than 100 ka (Hein et al., 2009; 2011; 2017; Darvill et al., 2015; Mendelova et al., 2020; Leger et al., 2023), the Arroyo Verde moraines could correspond to MIS 6 or MIS 8. This extends our interpretation to include an older advance and highlights an avenue for future research. This change is reflected throughout the paper, including in the abstract, conclusion, and Figure 6.

Minor suggestions

Line 71: "a relative ages"; there's a plural mismatch.

Done.

Line 73: Can you specify which you mean by "these moraines", you listed four systems, but I think you are just referring to the Puerto Banderas at the end of the preceding sentence.

We moved that sentence to the end of the last paragraph to improve readability as follows: **Lastly, Strelin and Malagnino (1996, 2009) suggested that the Lago Argentino glacier lobe readvanced and deposited the Puerto Banderas moraines westwards of El Tranquilo moraines. The latter are known to date from the Antarctic Cold Reversal, deposited about 10 km from the modern ice from at ~13,000 cal yrs before present (Strelin et al., 2011).**

Line 75: I think you mean Caldenius' framework, not Strelin (the last work you cited). I suggest moving the final sentence in Line 74 to the end of the next paragraph (e.g. Line 81) so the chronology of work is in order.

Done. We modified parts of this section to improve the narrative flow so that the chronology of the work is in order as stated in the comment above.

Line 86: "Last two glaciations" is ambiguous (in terms of them/you naming them? Or in terms of timing?) You can just state that you only focus on the Arroyo Verde and El Tranquilo moraines, or provide their stratigraphic order.

We agree with the reviewer, and we modify the text to simplify the flow of the paragraph: **For this work, we only focus on the Arroyo Verde and El Tranquilo moraines, as identified by Strelin and Malagnino (1996) (Fig.1C).**

Line 133: I'm not sure this scans quite right: I think you are deciding to report with zero erosion given ages overlap within uncertainties and doing so does not alter your main findings. Perhaps adjust the wording here because I do not think you can assume the erosion rate was zero based on the evidence provided.

Wording was modified to make it clear that despite erosion rates being non-zero, we still report our ages with an erosion rate equivalent to zero, given that using different erosion rates does not alter the main implications of this work. Modified as follows: **Since the outcomes of using different erosion rates (Table S1, Fig. S6) do not change the main results of this work and given that age differences overlap within analytical uncertainties, we use an erosion rate equivalent to zero for all the samples for our reported ages and interpretations**.

Line 144: Blomdin et al. used IRSL in southern Patagonia; perhaps a useful reference here (Blomdin, R., Murray, A., Thomsen, K.J., Buylaert, J.P., Sohbati, R., Jansson, K.N. and Alexanderson, H., 2012. Timing of the deglaciation in southern Patagonia: Testing the applicability of K-Feldspar IRSL. Quaternary Geochronology, 10, pp.264-272.)

We thank the reviewer for pointing out this study. We added this reference to the text and modified it accordingly: **Although OSL dating of quartz has been performed in Patagonia (Smedley et al., 2016), lithologies local to the Río Santa Cruz contain little quartz. Therefore, we apply IRSL dating of feldspar grains, as previously performed in other glacial settings within southern Patagonia (Blomdin et al., 2012).**

Line 166: Should "by moraine ridges" be "of moraine ridges"?

Noted and modified.

Line 182: Figure 3: Please include photos of all sampled boulders, if possible. It is useful for assessing what was sampled, and for anyone wanting to revisit sample sites in the future.

Figure 3 includes photos of boulders sampled for El Tranquilo I and Arroyo Verde moraines. On the other hand, Figure 4 includes field photos of sampled boulders from El Tranquilo II moraines.

Line 191: Point to Figure S2 again.

Done.

Line 188: Should be "inboard of", I think.

Noted and modified.

Line 194: "by in"; should be one or the other.

Noted and modified as follows: **by Strelin and Malagnino (1996).**

Line 203: "glacial transgressive facies" implies overlapping / overriding glacial sedimentology to me, but do you mean glaciofluvial, like the rest of the section? Perhaps you are referring to moraines overriding glaciofluvial features? In which case, start with the glaciofluvial features about which this section focusses and then explain that sorm landforms are indicative of glacial transgression.

We removed the lines: 'glacial transgressive facies', since they belong to the glaciofluvial section, and it confuses the narrative flow here.

Line 211: Why are meltwater channels not descrived in Section 4.1.3 on Glaciofluvial features?

Done and moved to glaciofluvial.

Line 213: Why are palaeo(lacustrine) features mentioned here again, rather than just in Section 4.1.4 on Glaciolacustrine features?

Done and moved to glaciolacustrine.

Line 249: The rationale for excluding the oldest age here and not the youngest is unclear. With only four ages, I question whether any samples should be removed at all. See broader point about these older ages in the major revisions section. There is a good argument that removing outliers should be done with great caution, or that all ages should be kept in and metrics such as the median be used instead (outlined in Dortch, J.M., Tomkins, M.D., Saha, S., Murari, M.K., Schoenbohm, L.M. and Curl, D., 2022. A tool for the ages: the probabilistic cosmogenic age analysis tool (P-CAAT). Quaternary geochronology, 71, p.101323.) Additionally, there is a lot of work on understanding age distributions from these sorts of environments in Patagonia that would counter you rejecting older ages as inherited (Kaplan, M.R., Douglass, D.C., Singer, B.S., Ackert, R.P. and Caffee, M.W., 2005. Cosmogenic nuclide chronology of pre-last glacial maximum moraines at Lago Buenos Aires, 46 S, Argentina. Quaternary Research, 63(3), pp.301-315.; Hein, A.S., Cogez, A., Darvill, C.M., Mendelova, M., Kaplan, M.R., Herman, F., Dunai, T.J., Norton, K., Xu, S., Christl, M. and Rodés, Á., 2017. Regional mid-Pleistocene glaciation in central Patagonia. Quaternary Science Reviews, 164, pp.77-94.)

We agree with the reviewer and consider all the ages for the oldest landform for our calculations, without excluding any result. We report the mean age and the standard deviation of this calculation, and we modify the interpretations accordingly. Additionally, we report the weighted mean and weighted standard deviation in Table 2.

Line 250: Correspondance with MIS 6 needs to be much more speculative here.

We agree, and we modify this section following advice from both reviewers. For more information, please refer to the modified interpretation of the Arroyo Verde moraines.

Line 258: The wording does not quite work here; needs slight editing.

Done and modified as follows: **Although our $^{10}$Be ages indicate that the El Tranquilo I and II moraines are statistically indistinguishable, our mapping indicates that they are stratigraphically distinct, and our dataset indicates that these moraines were occupied and abandoned during MIS 3 (57 – 29 ka; Lisiecki and Raymo, 2005), and that this advance constitutes the largest extent of the Argentino glacier lobe during the last glacial cycle.**

Line 250: Here and elsewhere (so perhaps state if it applies to all), explain what constitutes ther uncertainty here, and if it is weighted or not.

In the text, we report the ages for each moraine complex as the mean and the associated standard deviation. Additionally, we report the weighted mean and weighted standard deviation in Table 2. For this review, we clarify this in the text, for example, in Table 2 caption, as follows: **We report mean moraine age and standard deviation (bold), the weighted mean and standard deviation, and outliers (*).**

Line 260: I think the word "preserved" needs inserting, as it's possible (though unlikely) that moraines recording a more extensive advance were subsequently destroyed by outwash from the El Tranquilo advance in such a topographically-constrained system.

We agree with the reviewer, and we added the word preserved to Line 260 to note that a more extensive advance could have been overprinted by the MIS 3 advance and the one we dated is the largest preserved advance of the Lago Argentino lobe during the last glacial cycle. We modified the text accordingly: **…and this advance constitutes the largest preserved extent of the Argentino glacier lobe during the last glacial cycle**. Additionally, we add more text in line 309 (section 5.4.1 Comparison with other glacial records during Marine Isotope Stage 3 [MIS 3]) the following: **We determine that this outlet lobe reached its maximum extent during MIS 3, when it deposited the El Tranquilo I and II moraines ~120 km away from the modern ice front, constituting the earliest and largest preserved advance of the last glacial cycle.**

Line 262: Here, please give the raw mean age and associated error before moving on to explain why you are removing some ages and providing an updated mean and error.

Done. We added the mean age and the standard deviation of all ages before outlier removal.

Line 265: "the valley center".

Done.

Line 265: Small boulders do not inherently indicate exhumation, but they might be more susceptible to it if it occurred. Alter framing to reflect this here.

Done. We modified the text as follows: **For instance, boulders around valley-center rarely exceed half a meter in height, making these smaller boulders more susceptible to exhumation that could result in younger ages.**

Line 268: Be clear throughout about number of samples. E.g. here "we exclude three ages".

Addressed.

Line 280: Figure 6: This should include all ages, so extend back to >243 ka. The moraine ages are rather selective here. A classic of the region—Lago Buenos Aires—is missing (e.g. Kaplan's work, and Hein et al., 2017), as is Hein's work in Lago Pueyrredon. Their chronologies are highly relevant for this work.

We modified figure 6 to include additional relevant studies to support our interpretations, such as Lago Buenos Aires and Lago Pueyrredón.

Line 281: I feel this section needs to better reflect uncertainties in the new chronology; see major revisions.

Agreed. We modified this section and expanded our discussion to include the paleoclimate implications of our record from MIS 6 through MIS 8.

Line 300: Not clear why this is a separate section; it could be added to the start of Section 5.4.1

Addressed and modified.

Line 323: "a millennial-scale moraine chronology"?

Done.

Line 359: Here and elsewhere, worth reflecting that Davies et al. looked at 35 ka to present, so were not necessarily saying maxima occurred then, just that advances were not exceeded after that point.

We acknowledge that the PATICE reconstruction does not imply that the Patagonian Ice Sheet reached maximum extent at 35 ka, therefore we cite other studies involving glacial advances that occurred during MIS 3. We modified the sentence in Line 359 accordingly, and we refer the reader to section 5.4.1 'Comparison with other glacial records during Marine Isotope Stage 3 (MIS 3)', to make our point clearer.

Line 361: Mendelova et al. (2020) talked about the role of ice masses changing moisture patterns and should be referenced here (Mendelová, M., Hein, A.S., Rodés, Á. and Xu, S., 2020. Extensive mountain glaciation in central Patagonia during Marine Isotope Stage 5. Quaternary Science Reviews, 227, p.105996.).

Done and added to line 365.

Line 416: This does not sound right. There is evidence for glacial activity (retreating from moraines) during the Holocene (e.g. Sagredo et al., 2021; Reynhout et al., 2019; Strelin et al., 2014; Kaplan et al., 2016; Garcia et al., 2020). See Briner & Darvill 2024 for reference (https://doi.org/10.1016/B978-0-323-99931-1.00198-7). You cite some of these papers, so I wonder if it is just unclear what you mean here.

We agree with the reviewer that the last sentence of the penultimate paragraph as well as the last sentence of the last paragraph would benefit from reorganization. We note that during the Holocene, several glacial advances are recognized in Patagonia (Sagredo et al., 2021; Reynhout et al., 2019, 2022; Strelin et al., 2014; Kaplan et al., 2016; Garcia et al., 2020, Hall et al., 2020). We emphasize that our aeolian record coincides with periods of drier and warmer conditions that agree with the paleolimnological and paleoecological record from Lago Cipreses (Moreno et al., 2014; 2018). Additionally, this record coincides with a paucity of glacial advances during these drier/warmer periods. We modified the text accordingly: **While evidence for glacial growth is available from Southern Patagonia during the Holocene (Strelin et al., 2014; Kaplan et al., 2016; Reynhout et al., 2019; Sagredo et al., 2021; García et al., 2020), the lack of evidence for glacial advances at ~10.8 ka and ~3.6 ka coincides with these warm and dry periods defined by pollen assemblages (Moreno et al., 2018).**